# Dual Mean-Teacher: An Unbiased Semi-Supervised Framework for Audio-Visual Source Localization

**Yuxin Guo**[1,2,3]**, Shijie Ma**[1,2]**, Hu Su**[1,2]**, Zhiqing Wang**[1,2]**, Yuhao Zhao**[1,2]**, Wei Zou**[1,2]*

Siyang Sun[3], Yun Zheng[3]

[1]School of Artificial Intelligence, University of Chinese Academy of Sciences, Beijing, China

[2]State Key Laboratory of Multimodal Artificial Intelligence Systems (MAIS),
Institute of Automation of Chinese Academy of Sciences, Beijing, China

[3]DAMO Academy, Alibaba Group

{guoyuxin2021, wei.zou}@ia.ac.cn

## Abstract

Audio-Visual Source Localization (AVSL) aims to locate sounding objects within video frames given the paired audio clips. Existing methods predominantly rely on self-supervised contrastive learning of audio-visual correspondence. Without any bounding-box annotations, they struggle to achieve precise localization, especially for small objects, and suffer from blurry boundaries and false positives. Moreover, the naive semi-supervised method is poor in fully leveraging the information of abundant unlabeled data. In this paper, we propose a novel semi-supervised learning framework for AVSL, namely **Dual Mean-Teacher (DMT)**, comprising two teacher-student structures to circumvent the *confirmation bias* issue. Specifically, two teachers, pre-trained on limited labeled data, are employed to filter out noisy samples via the consensus between their predictions, and then generate high-quality pseudo-labels by intersecting their confidence maps. The sufficient utilization of both labeled and unlabeled data and the proposed unbiased framework enable DMT to outperform current state-of-the-art methods by a large margin, with CIoU of **90.4%** and **48.8%** on Flickr-SoundNet and VGG-Sound Source, obtaining **8.9%**, **9.6%** and **4.6%**, **6.4%** improvements over self- and semi-supervised methods respectively, given only $< 3\%$ positional-annotations. We also extend our framework to some existing AVSL methods and consistently boost their performance. Our code is available at `https://github.com/gyx-gloria/DMT`.

## 1 Introduction

Visual and auditory perception is crucial for observing the world. When we hear a sound, our brain will extract semantic information and locate the sounding source. In this work, we focus on Audio-Visual Source Localization (AVSL) [1, 2], with the purpose of accurately locating sounding objects in frames based on their paired audio clips. Beyond this scope, AVSL also plays a crucial role in many downstream tasks including environmental perception [3], navigation [4, 5], sound separation [6, 7] and event localization [8]. Therefore, accurate localization is of utmost importance.

In the literature of AVSL [9, 10, 11], the conventional paradigm is to employ self-supervised contrastive learning based on audio-visual correspondence. However, most of them suffer from some serious challenges. From the performance perspective, there are issues such as blurry boundaries, inability to converge to specific objects, and the predicted sounding regions that are too large to accurately locate objects, especially *small objects*. In terms of the learning stage, a single model alone is unable to recognize and filter out *false positives*, *i.e.*, noisy samples with no visible sounding sources, which could affect the entire learning process of the model.

---

*Corresponding author.

37th Conference on Neural Information Processing Systems (NeurIPS 2023).

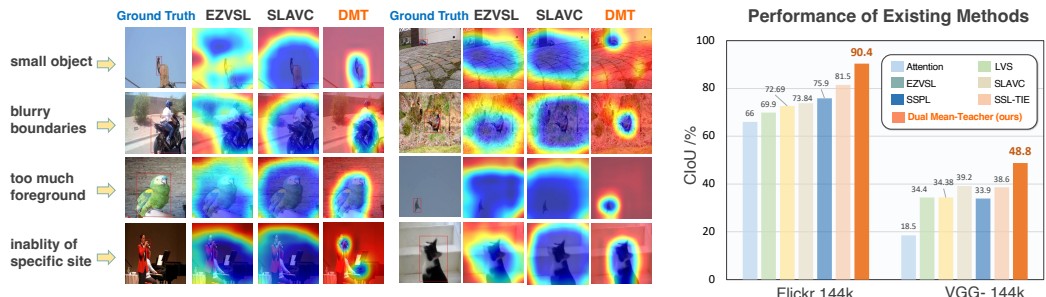

Figure 1: Comparison of existing Audio-Visual Source Localization (AVSL) methods and the proposed Dual Mean-Teacher (DMT). **Left:** DMT has greatly addressed severe issues including inaccurate small object localization, blurry boundaries, and instability. **Right:** DMT outperforms previous by a large margin on Flickr and VGG-ss datasets.

In essence, AVSL is a dense prediction task, which can not be directly accomplished from a shared global image representation [12], requiring models to capture fine local features in order to accurately predict object locations, *i.e.*, achieving precise pixel-level localization is not feasible without positional annotations. Unluckily, the number of samples with location labels is extremely limited. As a result, we resort to Semi-Supervised Learning [13] (SSL) to fully leverage the labeled data.

Considering that self-supervised AVSL is not fully learnable, Attention10k [14, 15] extended the self-supervised model to an SSL model by directly appending a supervised loss on labeled data, which is the first semi-supervised attempt in the field. Nevertheless, simply leveraging labeled data might lead to overfitting and neglect to fully harness the underlying unlabeled data. Given these issues, we resort to pseudo-labeling [16]. However, directly introducing pseudo-labeling could lead to *confirmation bias* [17] which cannot be adequately rectified by a single model.

To tackle these challenges, we break away from traditional self-supervised learning and propose a more sophisticated Semi-Supervised Audio-Visual Source Localization (SS-AVSL) framework, called Dual Mean-Teacher (DMT), which adopts a double teacher-student structure in a two-stage training manner. We consider previous AVSL methods as a single student unit. To fully leverage positional annotations and training data, we extend it to a classic semi-supervised framework Mean-Teacher [18]. To address the issue of *confirmation bias*, we expand it into a dual independent teacher-student structure with designed modules of Noise Filtering, Intersections of Pseudo-Labels (IPL), as shown in Figure 2. Specifically, teachers are pre-trained on a limited amount of labeled data in Warm-Up stage, establishing a solid foundation, in the subsequent Unbiased-Learning Stage, dual teachers filter out noisy samples and rectify pseudo-labels. In more detail, the Noise Filtering module effectively rejects noise samples by leveraging consensus, *i.e.*, agreement, between dual teachers, ensuring high-quality training data, then IPL module generates precise pseudo-labels by intersecting the predictions from both teachers. DMT eliminates the influence of *confirmation bias* by rejecting noisy samples and improving the quality of pseudo-labels, which effectively tackles the issues of false positives and greatly improves localization performance.

In summary, our method contributes to the following three aspects. **Firstly,** we introduce a novel unbiased framework based on a pseudo-labeling mechanism for semi-supervised AVSL, which could maximize the utilization of both labeled and unlabeled data, effectively address the challenge of limited annotated data, and mitigate the issue of *confirmation bias*. **Moreover,** compared to existing approaches, DMT achieves much remarkable localization performance, with better small object localization and stronger generalization capability, which significantly elevate the performance of current methods. **Finally,** DMT can be summarized as a semi-supervised learning paradigm and could be combined with existing (weakly-supervised) AVSL methods to consistently boost their performance.

## 2   Related Works

**Semi-Supervised Learning.**   Semi-Supervised Learning (SSL) [13, 19] leverages a small amount of labeled data to unlock the potential of unlabeled data for better model learning. One line of work relies on consistency regularization [18, 20, 21] to encourage models to behave similarly under different perturbations. An orthogonal idea is to generate high-quality pseudo-labels [16, 22] on

unlabeled data to retrain models for better performance. The quality of pseudo-labels is crucial. Current methods [23, 24, 25] combine the above two paradigms to achieve remarkable results.

**Audio-Visual Source Localization.** The key to Audio-Visual Sound Localization (AVSL) endeavors to establish the correspondence between visual objects and their corresponding sounds by contrastive learning [26, 27]. Most existing methods predominantly utilize self-supervised or weakly-supervised approaches (for all of them employ pre-trained backbone). Some classical works, such as Attention10k [14, 15], DMC [2], LVS [28], EZVSL [9], SSPL [29], SSL-TIE [30] achieve improving performance over time. Other methods like DSOL [31], CoarsetoFine [32], mix and localize [33], and AVGN [11] pay attention to multi-source localization. In addition, some studies also address the issue of false positives and false negatives in self-supervised learning. For example, SLAVC [10] focuses on false positives and effectively overcomes the overfitting issue. IER [34] proposes a label-free method that targets the suppression of non-sounding objects, while Robust [35] considers both false positive and false negative issues. AVID [36] detects false negatives by defining sets of positive and negative samples via cross-modal agreement. Source separation [37, 38] and generative models [39] also achieve good results. However, most AVSL methods exhibit subpar performance in the absence of annotation.

**Semi-Supervised Learning in Localization.** Semi-Supervised Object Detection (SSOD) is one of the few applications of SSL in the localization field. The majority of SSOD methods, such as [40, 41, 42], utilize pseudo-labeling to enhance the localization performance. Moreover, some works like [43, 44] focus on the *confirmation bias* in SSOD. Similar to object detection, AVSL is a pixel-wise dense prediction task, heavily reliant on high-quality pseudo-labels. Attention10k [14, 15] is the first SS-AVSL work. It extends a self-supervised model to a semi-supervised framework by simply adding a supervised loss, aiming at fixing the false conclusions generated by weakly-supervised methods. However, this naive method may lead to overfitting and neglects the full utilization of unlabeled data. In contrast, we introduce a novel SS-AVSL framework based on pseudo-label mechanism, which can address *confirmation bias* and maximize the utilization of both labeled and unlabeled data, to achieve stronger localization performance.

## 3 Background

**Problem Definition.** Audio-Visual Source Localization (AVSL) aims to accurately locate the sound source within a given visual scene. We denote audio-visual pairs as $(a_i, v_i)$, where $a_i$ and $v_i$ represent the audio and visual modality, respectively. The objective is to generate a pixel-wise confidence map $\mathcal{P}$ indicating the location of the sound source.

**Contrastive Learning in AVSL.** Self-supervised AVSL methods commonly leverage audio-visual correspondence to maximize the similarity between frames and their corresponding audio clips (positive pairs) while minimizing the similarity among unpaired ones (negative pairs):

$$\mathcal{L}_{\text{unsup}} = -\mathbb{E}_{(a_i, v_i) \sim \mathcal{D}_u} \left[ \log \frac{\exp(s(g(a_i), f(v_i))/\tau_t)}{\sum_{j=1}^n \exp\left(s\left(g(a_i), f(v_j)\right)/\tau_t\right)} + \log \frac{\exp(s(f(v_i), g(a_i))/\tau_t)}{\sum_{j=1}^n \exp\left(s\left(f(v_i), g(a_j)\right)/\tau_t\right)} \right]. \tag{1}$$

where $\mathcal{D}_u$ are unlabeled datasets, $g(\cdot)$ and $f(\cdot)$ are audio and visual feature extractors. $\tau_t$ is the temperature coefficient. $s(\cdot)$ is consistency matching criterion. The predicted map $\mathcal{P}_i$ is typically calculated with cosine similarity $\text{sim}(\cdot)$ to represent the confidence of the presence of sounding objects:

$$\mathcal{P}_i = \text{sim}(g(a_i), f(v_i)) = \frac{\langle g(a_i), f(v_i) \rangle}{\|g(a_i)\| \cdot \|f(v_i)\|}. \tag{2}$$

**Learning with (Pseudo) Labels in AVSL.** When labeled data are available, one could apply supervised loss directly to learn to localize:

$$\mathcal{L}_{\text{sup}} = \mathbb{E}_{i \sim \mathcal{D}} H(\mathcal{G}_i, \mathcal{P}_i). \tag{3}$$

where $\mathcal{G}_i$ could be the ground truth or generated pseudo-labels, both $\mathcal{G}_i$ and $\mathcal{P}_i$ are in the form of binary confidence map. $H(\cdot, \cdot)$ is cross-entropy function across the two-dimensional spatial axes.

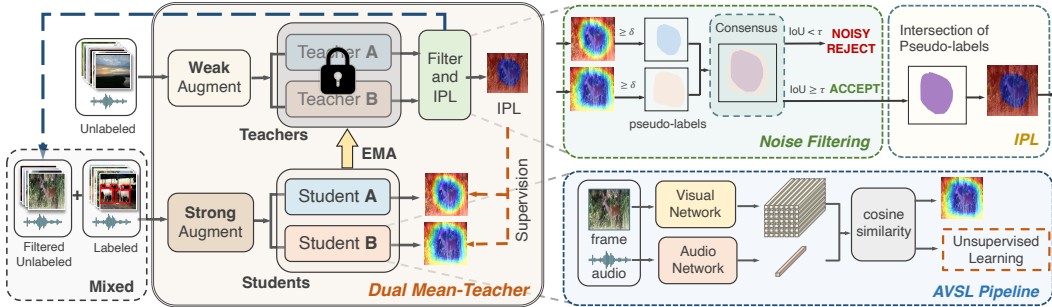

Figure 2: Overview of the proposed Dual Mean-Teacher framework. **Left:** Overall learning process of dual teacher-student pairs, two students are guided by both ground-truth labeled data and filtered unlabeled data with the Intersection of Pseudo-Labels (IPL). **Upper-right:** Details of Noise Filtering and IPL. Dual teachers reject noise samples based on their consensus and generate pseudo-labels on filtered data. **Lower-right:** Details of AVSL pipeline. Students are learned through contrastive learning and predict confidence maps for supervised learning with (pseudo) labels.

## 4 Dual Mean-Teacher

**Overview.** In this section, we mainly describe Dual Mean-Teacher (DMT) in the order of the learning process. Specifically, in the Warm-Up Stage (Section 4.1), two teachers are pre-trained on bounding-box annotated data to obtain a stable initialization. In the subsequent Unbiased-Learning Stage (Section 4.2), the Noise Filtering module and Intersection of Pseudo-Label (IPL) module collectively filter out noisy samples and generate high-quality pseudo-labels by two teachers to guide students' training, teachers are in turn updated by exponential moving average (EMA) of students.

From a unified perspective, existing AVSL methods can be viewed as a single student, which is later expanded into the semi-supervised classical framework Mean-Teacher [18], in order to fully utilize limited labeled data. To effectively address the *confirmation bias* issue, we further extend it to a double teacher-student framework, as shown in Figure 2. DMT adopts a teacher-student architecture, where each teacher and student contains two independent AVSL pipelines for audio-visual contrastive learning and generating localization results. Teachers provide students with stable unlabeled samples after Noise Filtering and their generated IPL. Students pass back the parameters to the teachers.

**General Notations.** Cross-entropy function $H(\cdot, \cdot)$ and two feature extractors $f(\cdot)$ and $g(\cdot)$ are already discussed in Section 3. By default, subscript $i$ indicates the $i$-th sample, while superscript $A$, $B$ denote two teacher-student pairs with $t$ and $s$ indicating teacher and student, respectively. $\mathcal{D}_l$ and $\mathcal{D}_u$ are labeled and unlabeled datasets. $\mathcal{G}_i$ and $\mathcal{P}_i$ are ground-truth and predicted confidence maps, both with size of $H \times W$. We apply strong $\mathcal{A}(\cdot)$ or weak $\alpha(\cdot)$ augmentation on visual inputs.

### 4.1 Warm-Up Stage

The quality of pseudo-labels is crucial to SSL, especially for localization tasks. Before generating pseudo-labels, we first pre-train dual teachers with bounding-box annotated data to achieve preliminary localization performance. In order to avoid overfitting, we apply strong augmentation and get augmented labeled dataset $\mathcal{D}_l = \{(a_i, \mathcal{A}(v_i), \mathcal{G}_i\}$. After extracting visual features $f^t(\mathcal{A}(v_i))$ and auditory features $g^t(a_i)$, the predicted map $\mathcal{P}_i^t$ can be obtained by Eq. (2). Then we utilize bounding-box annotations $\mathcal{G}_i$ as supervision:

$$\mathcal{L}_{\text{Warm-Up}} = \mathbb{E}_{(a_i, v_i) \sim \mathcal{D}_l} H(\mathcal{G}_i, \mathcal{P}_i^t). \tag{4}$$

### 4.2 Unbiased-Learning Stage

**Noise Filtering.** To mitigate *confirmation bias*, it is crucial to filter out noisy samples that are more likely to be false positives. As depicted in Figure 2, two predicted maps of the same sample are generated by dual teachers. It is clear that samples with higher reliability can be identified when the two predicted maps exhibit higher similarity, *i.e.*, there is more agreement and consensus between dual teachers, then the sample is reserved for pseudo-labelling. Conversely, when there is a significant discrepancy between the two maps, the sample will be considered as a false positive, such as frames without distinguishable sound objects or sounds that cannot be accurately represented by a bounding box (*e.g.*, wind sounds), such samples are rejected and discarded.

**Intersection of Pseudo-Labels (IPL).** By intersecting the foreground regions of two predicted maps on the filtered samples, one can generate positional pseudo-labels, named Intersection of Pseudo-Labels (IPL), to guide students' learning. With the pre-defined foreground threshold $\delta$, two predicted maps $\mathcal{P}_i^{t,A}, \mathcal{P}_i^{t,B}$ could be transferred to binary maps $\mathcal{M}_i^{t,A}$ and $\mathcal{M}_i^{t,B}$. Weak augmentation $\alpha(\cdot)$ is employed for teachers to generate high-quality pseudo-labels:

$$\mathcal{P}_i^t = \text{sim}(g^t(a_i), f^t(\alpha(v_i))), \tag{5}$$

$$\mathcal{M}_i^t = \mathbb{1}(\mathcal{P}_i^t \geq \delta). \tag{6}$$

We adopt the Intersection over Union (IoU) metric to quantify the similarity between the two maps $\mathcal{M}_i^{t,A}, \mathcal{M}_i^{t,B}$. If the IoU score exceeds the threshold $\tau$, the sample will be accepted, and the intersection of those two maps will be generated as its pseudo-label (IPL). Otherwise, it will be filtered out as a noise sample.

$$\mathcal{D}_u' = \left\{ (a_i, v_i) \mid \text{IoU}(\mathcal{M}_i^{t,A}, \mathcal{M}_i^{t,B}) \geq \tau, \, \forall (a_i, v_i) \in \mathcal{D}_u \right\}. \tag{7}$$

$$\mathcal{IPL}(a_i, v_i) = \mathcal{M}_i^{t,A} \cdot \mathcal{M}_i^{t,B}. \tag{8}$$

The newly selected unlabeled dataset is applied to the student model along with the corresponding high-quality IPL.

**Students Learning without bias.** To suppress *confirmation bias* more effectively, we mix labeled and new unlabeled datasets. Both ground-truth annotations and high-quality IPL are employed to train the student models:

$$\mathcal{D}_{mix} = \mathcal{D}_l \cup \mathcal{D}_u' = \{(a_i, v_i), \widehat{\mathcal{G}}_i\}, \text{ where } \widehat{\mathcal{G}}_i = \begin{cases} \mathcal{G}_i & \text{if } (a_i, v_i) \in \mathcal{D}_l \\ \mathcal{IPL}(a_i, v_i) & \text{if } (a_i, v_i) \in \mathcal{D}_u' \end{cases} \tag{9}$$

In addition, we incorporate consistency regularization [45] in the semi-supervised learning process. Specifically, for a given sample, we obtain IPL from the teachers on weakly augmented images while strong augmentations are applied for samples of students. By enforcing consistency between IPL and students' predictions, DMT could be more stable with better generalization ability.

$$\mathcal{P}_i^s = \text{sim}(g^s(a_i), f^s(\mathcal{A}(v_i))), \tag{10}$$

$$\mathcal{L}_{\text{sup}} = \mathbb{E}_{i \sim \mathcal{D}_{mix}} H(\widehat{\mathcal{G}}_i, \mathcal{P}_i^s). \tag{11}$$

Similar to the AVSL method mentioned in Section 3, students are also trained by audio-visual correspondence of contrastive learning loss. Here, we introduce an attention module to add attention to the sounding region in the frame:

$$f_{att}(v_i) = \frac{\exp\left(\mathcal{P}_i(x, y)\right)}{\sum_{x,y} \exp\left(\mathcal{P}_i(x, y)\right)} \cdot f(v_i). \tag{12}$$

Then, the full semi-supervised loss could be derived with $\mathcal{L}_{\text{sup}}$ (see Eq. (11)) on $\mathcal{D}_{mix}$ and $\mathcal{L}_{\text{unsup}}$ (see Eq. (1)) on $\mathcal{D}_u$:

$$\mathcal{L}_{\text{full}} = \left(\mathcal{L}_{\text{sup}}^A + \mathcal{L}_{\text{sup}}^B\right) + \lambda_u \left(\mathcal{L}_{\text{unsup}}^A + \mathcal{L}_{\text{unsup}}^B\right). \tag{13}$$

**Update of Students and Teachers.** Students are updated via gradient descent of $\mathcal{L}_{\text{full}}$, while dual teachers are updated through the exponential moving average (EMA) of corresponding students:

$$\theta_m^s \leftarrow \theta_{m-1}^s - \gamma \frac{\partial \mathcal{L}_{\text{full}}}{\partial \theta_{m-1}^s}, \quad \theta_m^t \leftarrow \beta \theta_{m-1}^t + (1 - \beta)\theta_m^s. \tag{14}$$

The slowly progressing teachers can be regarded as the ensemble of students in recent training iterations, which enables stable progress in training.

### 4.3 Unbiased Superiority of Dual Mean-Teacher

For dense prediction tasks such as AVSL, employing pseudo-labels for model training can easily accumulate errors and lead to sub-optimal results. In our DMT framework, The unbiased characteristics could be attributed to the following three factors: (i). Noise Filtering ensures that only stable samples are utilized to train. (ii). IPL generates high-quality pseudo-labels. (iii). Pre-train dual teachers on bounding-box annotated data with strong augmentation in Warm-Up Stage. The above conclusion will be validated in subsequent ablation studies in Section 5.4.

Table 1: Comparison results on Flickr-SoundNet. Models are trained on Flickr 10k and 144k. † indicates our reproduced results, others are borrowed from original papers. Attention10k-SSL is of 2k labeled data supervision. We report the proposed DMT results from both stages as `stage-2(stage-1)`. $|\mathcal{D}_l|$ denotes the number of labeled data.

| Methods | Flickr 10k | | Flickr 144k | |
|---|---|---|---|---|
| | CIoU | AUC | CIoU | AUC |
| Attention10k [14, 15] | 43.60 | 44.90 | 66.00 | 55.80 |
| CoarsetoFine [32] | 52.20 | 49.60 | – | – |
| DMC [2] | – | – | 67.10 | 56.80 |
| LVS [28] | 58.20 | 52.50 | 69.90 | 57.30 |
| EZVSL [9] | 62.65 | 54.89 | 72.69 | 58.70 |
| SLAVC† [10] | 66.80 | 56.30 | 73.84 | 58.98 |
| SSPL [29] | 74.30 | 58.70 | 75.90 | 61.00 |
| SSL-TIE† [30] | 75.50 | 58.80 | 81.50 | 61.10 |
| Attention10k-SSL [14, 15] | 82.40 | 61.40 | 83.80 | 61.72 |
| Ours ($|\mathcal{D}_l| = 256$) | 87.20 (84.40) | 65.77 (59.60) | 87.60 (84.40) | 66.28 (59.60) |
| Ours ($|\mathcal{D}_l| = 2k$) | 87.80 (85.60) | 66.20 (63.18) | 88.20 (85.60) | 66.63 (63.18) |
| Ours ($|\mathcal{D}_l| = 4k$) | **88.80** (86.20) | **67.81** (65.56) | **90.40** (86.20) | **69.36** (65.56) |

# 5 Experiments

With limited annotated data, DMT could significantly raise the performance of AVSL and address the severe issues, *e.g.*, false positives and poor localization ability on small objects. Then, we direct our focus towards answering the following questions with ablation experiments 5.4:

- What is the individual contribution of each module to the performance gains?

- How does annotation enhance localization performance?

- Why can DMT outperform the existing semi-supervised AVSL method?

- Is it necessary to warm up dual teachers?

- How to effectively mitigate *confirmation bias* in AVSL?

## 5.1 Experimental Settings

**Datasets.** We conduct experiments on two large-scale audio-visual datasets: Flickr-SoundNet [14, 15] and VGG Sound Source [46], where there are 5,000 and 5,158 bounding-box annotated samples, respectively. For labeled data, we randomly select 4,250 for training, 500 for validating, and keep the same test sets with 250 samples as previous works [9, 10, 28, 30]. Moreover, we select a subset of 10k and 144k unlabeled samples to train as before. Details can be found in Appendix B.1.

**Audio and Visual Backbones.** For visual backbones, we followed prior work and used ResNet-18 [47] pre-trained on ImageNet [48]. For audio backbones, we select the pre-trained VGGish [49] and SoundNet [50] with semantic audio information. Further details can be found in Appendix B.2.

**Metrics.** We report the Consensus Intersection over Union (CIoU) and Area Under Curve (AUC), following previous settings [14, 15]. CIoU represents the localization accuracy, samples with IoU above the threshold $\delta = 0.5$ are considered to be accurately located. Considering small objects, we introduce Mean Square Error (MSE), which measures the average pixel-wise difference between two maps without binarization, making it more suitable for evaluating dense prediction tasks on small objects. More details are shown in Appendix B.3.

**Implementation details.** For audio clips, we pass $96 \times 64$ log-mel spectrograms to VGGish, and the output is a 512D vector, while the raw waveform of the original 3s audio clip is sent to SoundNet. For frames, we used an input image of size $256 \times 256 \times 3$, with $224 \times 224 \times 512$ as output. We choose RandAug [51] as strong augmentation, while random cropping, resizing, and random horizontal flip as weak augmentation. We set $\delta$ as 0.6 and $\tau$ as 0.7. More experiments of hyperparameters are shown in Appendix C.5.

Table 2: Comparison results on VGG-ss. Models are trained on VGG-Sound 10k and 144k.

| Methods | VGG-Sound 10k | | VGG-Sound 144k | |
|---|---|---|---|---|
| | CIoU | AUC | CIoU | AUC |
| Attention10k [14, 15] | 16.00 | 28.30 | 18.50 | 30.20 |
| LVS [28] | 27.70 | 34.90 | 34.40 | 38.20 |
| EZVSL [9] | 32.30 | 33.68 | 34.38 | 37.70 |
| SLAVC† [10] | 37.80 | 39.48 | 39.20 | 39.46 |
| SSPL [29] | 31.40 | 36.90 | 33.90 | 38.00 |
| SSL-TIE† [30] | 36.80 | 37.21 | 38.60 | 39.60 |
| Attention10k-SSL† [14, 15] | 38.60 | 38.26 | 39.20 | 38.52 |
| Ours ($|\mathcal{D}_l| = 256$) | 41.20 (39.40) | 40.68 (38.70) | 43.60 (39.40) | 41.88 (38.70) |
| Ours ($|\mathcal{D}_l| = 2k$) | 43.20 (42.60) | 40.82 (40.75) | 45.60 (42.60) | 43.24 (40.75) |
| Ours ($|\mathcal{D}_l| = 4k$) | **46.80** (43.80) | **43.18** (41.63) | **48.80** (43.80) | **45.76** (41.63) |

## 5.2 Comparison with the State-of-the-art Methods

Comprehensive experiments show that DMT achieves the state-of-the-art performance among all existing methods on both datasets, and showcases several advantages.

**Effective Utilization of Finite Annotations and Remarkable Performance.** We tested DMT's localization performance with varying amounts of labeled data and found that it consistently outperforms state-of-the-art methods when with 256, 2k, and 4k labeled data. Notably, even with just 256 labeled data, DMT achieved an accuracy of $87.2\%$ to $87.6\%$, showing a significant improvement in CIoU by around 10 absolute points compared to preceding models. Additionally, our model shows a $3\%$ absolute improvement in CIoU compared to a supervised-only model. Furthermore, DMT maintains superior performance in complex and open environments, as demonstrated in Table 2 and Table 3c, indicating strong generalization capabilities. These results highlight DMT's ability to improve localization performance by utilizing more unlabeled data.

Table 3: Performance comparisons in existing issues (small objects localization and false positives) and complex scenarios (open set). The results of small objects and open set are tested on the VGG-ss dataset, while false positives are reported on the Flickr dataset.

| Methods | Small Testset | | Medium Testset | |
|---|---|---|---|---|
| | MSE↓ | IoU↑ | MSE↓ | IoU↑ |
| LVS [28] | 0.515 | 0.021 | 0.441 | 0.265 |
| EZVSL [9] | 0.566 | 0.023 | 0.467 | 0.268 |
| SLAVC [10] | 0.705 | 0.021 | 0.568 | 0.220 |
| Ours | **0.160** | **0.025** | **0.174** | **0.335** |

(a) Small objects.

| Methods | AP↑ | max-F1↑ | Acc↑ |
|---|---|---|---|
| LVS [28] | 9.80 | 17.90 | 19.60 |
| DMC [2] | 25.56 | 41.80 | 52.80 |
| EZVSL [9] | 46.30 | 54.60 | 66.40 |
| SLAVC [10] | 51.63 | 59.10 | 83.60 |
| Ours | **53.56** | **62.80** | **91.60** |

(b) False positives.

| Methods | CIoU↑ | AUC↑ |
|---|---|---|
| LVS [28] | 26.30 | 34.70 |
| EZVSL [9] | 39.57 | 39.60 |
| SLAVC [10] | 38.92 | 41.17 |
| Ours | **43.12** | **42.81** |

(c) Open set.

**Significant Advancement in Small Subset Localization.** We categorize objects based on their bounding box pixel area into small ($1 \sim 32^2$), medium ($32^2 \sim 96^2$), large ($96^2 \sim 144^2$) and huge ($144^2 \sim 224^2$). We tested different methods on small and medium objects in the VGG-Sound dataset, focusing on the challenges of detecting small objects and reducing false positives mentioned earlier. The results in Table 3a show that DMT significantly improves performance, especially in terms of MSE metric. Despite some errors in the IoU metric, DMT still outperforms previous methods. The results in Figure 1 show that DMT accurately locates sounding objects with clear boundaries and precisely convergence to object contours, unlike previous methods that often have excessive or insufficient foreground regions, especially for small objects. These results demonstrate the effective and precise localization of small objects achieved by DMT. More experiments of different object sizes are in Appendix C.7.

**Capability of Learning Rich Semantic Information.** We present visualized predictions for testsets of varying sizes in Figure 1 (Left). It is evident that our approach achieves remarkable precision in localizing sounding sources. It accurately locates the position of sounding objects and precisely converges to their boundaries, while prior methods usually have excessive or insufficient foreground regions, particularly in the case of small objects.

Table 5: Main ablation study results. Models are trained on Flickr 144k and tested on Flickr-SoundNet testset, where 'S' and 'V' denote SoundNet and VGGish respectively.

| Modules | | | | | Performance | | |
| --- | --- | --- | --- | --- | --- | --- | --- |
| # Teachers | Backbone | Filter | IPL | EMA | CIoU↑ | AUC↑ | MSE↓ |
| One teacher | (a). S | ✗ | ✗ | ✗ | 80.20 | 53.57 | 0.458 |
| | (b). V | ✗ | ✗ | ✗ | 81.80 | 55.92 | 0.379 |
| Dual teachers | (c). S+V | ✗ | ✗ | ✗ | 82.20 | 55.16 | 0.382 |
| | (d). S+V | ✗ | ✗ | ✓ | 82.80 | 59.38 | 0.375 |
| | (e). S+V | ✗ | ✓ | ✗ | 83.60 | 62.83 | 0.359 |
| | (f). S+V | ✓ | ✗ | ✗ | 84.80 | 65.58 | 0.259 |
| | (g). S+V | ✓ | ✗ | ✓ | 86.20 | 66.56 | 0.260 |
| | (h). S+V | ✗ | ✓ | ✓ | 86.60 | 66.35 | 0.274 |
| | (i). S+V | ✓ | ✓ | ✗ | 88.60 | 66.68 | 0.260 |
| | (j). S+V | ✓ | ✓ | ✓ | **90.40** | **69.36** | **0.237** |

It is worth noting that our method can even find out sounding objects overlooked in the manual annotations. For instance, in Figure 1 (Left, 4-th row), the heatmap reveals the presence of a piano, which is omitted in the manual annotation process. Furthermore, we assessed the model's capability to identify false positives, signifying instances where sounding objects are occasionally not visually observable within the image (off-screen), as shown in 3b. This reflects the ability of Dual Mean-Teacher to extract audio semantic information and effectively localize multiple sounding objects within a scene, a feat that eludes other methods. We attribute this capability to the semantic alignment of audio-visual features achieved through the pre-trained VGGish and SoundNet backbone.

**Capacity for Cross-Domain Generalization and Multi-Source Localization.** We tested DMT's generalization across different domains and its ability to localize multiple objects. Models trained using VGG-ss 144k were directly evaluated on MUSIC-solo [37], MUSIC-duet [37], and MUSIC-synthetic datasets [31, 52]. Figure 3 demonstrates DMT's strong generalization performance in the music do-main, outperforming other method. As shown in Figure 3, the previous method struggles to accurately localize multi-

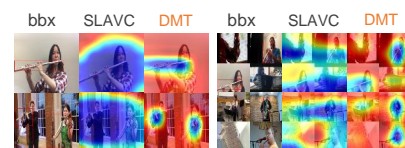

Figure 3: Performance on music-domain.

ple sounding objects, either missing them or including all sounding objects within a large foreground area. In contrast, DMT localizes each instrument accurately and separately. However, without category information for fine-grained training, it leads to sub-optimal performance in differentiating between multiple active and silent instruments. There is still significant room for improvement with multiple sounding objects and we plan to address this issue in future work.

## 5.3 Extensions of Dual Mean-Teacher

We replicated several existing methods and integrated them into our framework. Notably, the integration of the Dual Mean-Teacher showcases its ability to significantly enhance the performance of other existing methods. In Table 4, one can observe a noteworthy improvement in the CIoU of EZVSL from 62.65% to 87.20%, and SLAVC rising from 66.80% to 88.80%, which further reinforces the efficacy of our framework and highlight its flexible extensibility.

Table 4: Extension results of DMT with various audio backbones, with 'R', 'V' and 'S' denoting ResNet, VGGish and SoundNet.

| Methods | Backbones | CIoU↑ | AUC↑ | MSE↓ |
| --- | --- | --- | --- | --- |
| EZVSL w/o DMT | R | 62.65 | 54.89 | 0.428 |
| EZVSL w/ DMT | R+V | 85.30 | 65.80 | 0.312 |
| EZVSL w/ DMT | R+S | 85.95 | 66.12 | 0.298 |
| EZVSL w/ DMT | V+S | **87.20** | **67.74** | **0.256** |
| SLAVC w/o DMT | R | 66.80 | 56.30 | 0.386 |
| SLAVC w/ DMT | R+V | 86.10 | 66.24 | 0.288 |
| SLAVC w/ DMT | R+S | 86.30 | 66.58 | 0.283 |
| SLAVC w/ DMT | V+S | **88.80** | **68.69** | **0.247** |

## 5.4 Ablation Studies

**What is the individual contribution of each module to the performance gains?** In this section, we progressively analyze the performance gain from each module in detail. We choose a self-supervised approach as our baseline. Table 5 presents the results on the Flickr 144k training set with 10% annotated samples.

(i). Initially, we apply it under the semi-supervised pseudo-labeling mechanism, using only one backbone, as shown in Table 5(a) and Table 5(b). The localization performance improves with annotated data supervision, but the gain is limited due to poor-quality pseudo-labels.

(ii) Next, we extend it to a two-backbone architecture and sequentially introduce the filter, IPL, and EMA modules. The results demonstrate that all three modules contribute to performance improvement (3%, 1.8%, 1%). Notably, Filter shows the most significant impact on the model's improvement, for it effectively rejects noisy samples, ensuring the stability of the model.

(iii) Finally, we can observe that optimal performance is achieved through the joint integration of the three modules by effectively suppressing *confirmation bias*.

**How does annotation help localization?** We aim to demonstrate that even with extremely limited labeled data, significant performance can still be achieved. To this end, we investigated the performance of our model from $0.5\%$ to $10\%$, and report the results in Table 6. Our model consistently outperforms state-of-the-art approaches across all ratios. Furthermore, with a constant amount of unlabeled data, as the proportion of labeled data increases, our model's performance continues to improve, highlighting the significant impact of labeled data.

We investigated the impact of varying amounts of unlabeled data while keeping labeled data constant. Experimental results in Table 6 show that increasing the amount of unlabeled data improves localization performance, which seems contradictory to the previous conclusion about the proportion of labeled data, but actually, it demonstrates that labeled data can effectively leverage the unlabeled data. Based on our analysis, labeled data not only provides annotation information but also effectively enhances the power of unlabeled data, resulting in significant performance improvements.

Table 6: Performance on various labeled ratios % and multiple × on Flickr 144k.

| Labeled ratio % | CIoU | AUC |
|---|---|---|
| 0.5% (200/40k) | 84.80 | 63.58 |
| 1% (400/40k) | 86.20 | 65.16 |
| 2% (800/40k) | 87.20 | 65.94 |
| 5% (2k/40k) | 87.60 | 67.44 |
| 10% (4k/40k) | **88.40** | **68.12** |

| Multiple × | CIoU | AUC |
|---|---|---|
| 2.5× (4k/10k) | 88.00 | 67.80 |
| 5× (4k/20k) | 88.20 | 67.91 |
| 10× (4k/40k) | 88.40 | 68.12 |
| 20× (4k/80k) | 89.20 | 68.44 |
| 40× (4k/200k) | **91.20** | **71.36** |

**Why can DMT outperform the existing semi-supervised AVSL method?** Both naive SSL and DMT have utilized labeled and unlabeled data. However, a key distinction is that naive SSL employs unlabeled data only for contrastive loss, whereas DMT leverages pseudo-labels to incorporate unlabeled data into both contrastive loss and supervised loss, which amplifies the utilization of unlabeled data, thus enhancing generalization capability.

*Data Utilization.* We supplement the comparison experiments with fixed labeled data and an increase in unlabeled data from 10k to 200k, as shown in left part of 7. As the amount of unlabeled data increases, naive SSL exhibits only marginal improvement, whereas DMT shows more performance gains, indicating DMT can better use unlabeled data.

*Generalization Ability.* The right part of 7 highlights the limitations of naive SSL in the open set and in-the-wild datasets, suggesting that adding a supervised loss alone may lead to overfitting and weaken generalization. In contrast, DMT effectively leverages pseudo-label for improved generalization capability.

Table 7: Comparison of two SS-AVSL methods. * denotes the results from the original paper. 'sim-avsl' denotes the simple self-supervised AVSL model we use. We report the CIoU below.

| | 2.5k/10k | 2.5k/144k | 2.5k/200k | open set | cross-datasets |
|---|---|---|---|---|---|
| attention10k + naive SSL | 84.00*/83.68 | 84.40*/84.08 | 84.24 | 19.60 | 62.20 |
| attention10k + DMT (ours) | **88.00** | **89.52** | **90.40** | **42.64** | **87.26** |
| sim-avsl + naive SSL | 83.84 | 84.24 | 84.40 | 20.80 | 60.60 |
| sim-avsl + DMT (ours) | **88.24** | **89.76** | **91.12** | **43.10** | **89.80** |

**Is it necessary to warm up dual teachers?** We believe that the initialization of teachers and students is crucial, for the quality of pseudo-labels has a significant impact on model performance. To validate the effectiveness of this idea, we experimented to study the warm-up stage's impact on the model. We find that without the Warm-Up Stage, the model's improvement is very slow, and the performance eventually deteriorates. This indicates that without a good initialization, the model can

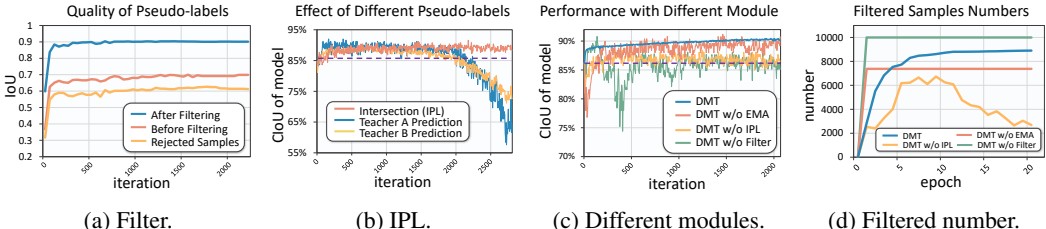

| (a) Filter. | (b) IPL. | (c) Different modules. | (d) Filtered number. |

Figure 5: The effect of each component (Noise Filtering, IPL and EMA) in DMT to suppress confirmation bias, together with the number of filtered samples for pseudo labeling depicted in (d).

accumulate errors, leading to *confirmation bias* issues. Therefore, we can conclude that Warm-Up is essential as it effectively suppresses *confirmation bias* in the early stages of training.

**How to effectively mitigate confirmation bias in AVSL?** In Section 4.3, we present the origins and mitigation strategies for *confirmation bias* in the localization task. In this section, we will demonstrate this. Figure 5a depicts the quality of pseudo labels before and after applying the filter module. It is evident that the quality of pseudo labels can be significantly improved after filtering. This observation highlights the effectiveness of the filter in eliminating noisy samples, which tend to be false-positive instances. Figure 5b depicts the comparison between using the direct outputs of each teacher as pseudo labels and utilizing their intersection, known as IPL. Where the purple dashed line represents the initialization value. The results clearly indicate that employing the direct outputs alone leads to the accumulation of bias, causing a deterioration in the quality of pseudo labels throughout the training process. Conversely, IPL consistently ensures the preservation of high-quality pseudo labels, thus mitigating the impact of bias.

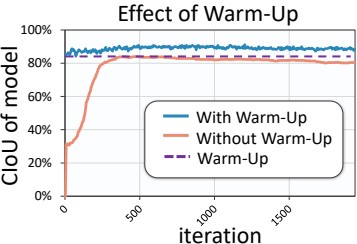

Figure 4: Effect of Warm-Up Stage.

Furthermore, Figure 5c visually presents the trend of model performance, revealing that the absence of any of these three modules results in a decline in model performance. However, we can see that EMA only affects the final performance of the model, and without the Filter module, the model's performance will be significantly affected by noise. Without the IPL module, the model will experience a continuous decline in performance due to erroneous estimation of pseudo-labels. Therefore, we find that the Noise Filtering module and IPL modules play a significant role in addressing the *confirmation bias* problem. Moreover, Figure 5d reflects that under the joint action of the three modules, DMT generates more accurate pseudo-labels and its performance continues to improve steadily.

## 6 Conclusion

In this paper, we advance the naive SS-AVSL work and propose a novel Semi-Supervised Audio-Visual Source Localization (SS-AVSL) framework, namely Dual Mean-Teacher (DMT), considering the importance of both limited annotated and abundant unlabeled data. From a unified perspective, existing self-supervised (weakly-supervised) AVSL methods could be referred to as a single student structure, while DMT employs dual teacher-student pairs to filter out noisy samples via the agreement of two teachers and generate high-quality pseudo labels to avoid *confirmation bias*. DMT has greatly enhanced AVSL performance and addressed intractable issues like false positives and inaccurate localization of tiny objects. Moreover, DMT is a learning paradigm and could be seamlessly incorporated into existing AVSL methods and consistently boost their performance.

We hope this work will bring more attention to SS-AVSL, provoke a reconsideration of pseudo-labeling, bias avoidance, and better utilization of the underlying unlabeled data, and thus stimulate more semi-supervised learning research in this dense prediction task.

## Acknowledgements

We would like to thank the National Natural Science Foundation of China under Grant 61773374 and the Major Basic Research Projects of Natural Science Foundation of Shandong Province under Grant ZR2019ZD07. We also appreciate Shuailei Ma, Kecheng Zheng, and Ziyi Wang for their valuable and insightful discussions.

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

# Appendix

**Brief Introduction.** The appendix is structured into four main sections: Algorithm, Experimental Settings, Supplementary Experiments, and Further Analysis. The main contents are as follows:

- A. **Algorithm:** Pseudo-codes and algorithm details.

- B. **Experimental Settings:** More detailed description of the datasets, backbones, metrics formula, implementation details and baselines.

- C. **Supplementary Experiments:** Results of cross-dataset evaluation, comparison of different predicted map, exploration of Warm-Up speed and its influence on the final results, false-positive rejection capability of Noise Filtering, investigation of hyperparameters for Filter, IPL, and EMA, effect of data augmentation and quality analysis (visualization).

- D. **Further Analysis:** Theoretical elaboration on the challenges faced by existing contrastive learning methods, and explanation of why contrastive learning alone cannot achieve precise localization.

## A  Algorithm

To make it more clear, Dual Mean-Teacher is specifically depicted in Algorithm 1.

---

**Algorithm 1** Dual Mean-Teacher algorithm.

---

1: **Input:** $\mathcal{D}_u = \{(a_i, v_i)\}$, $\mathcal{D}_l = \{(v_i, a_i), \mathcal{G}_i\}$ {labeled data and unlabeled data.}
2: **while** not reach the maximum iteration **do**
3:    **for** $(a_i, v_i)$ in $\mathcal{D}_u$ **do**
4:       **while** not reach the convergency of Warm-Up **do**
5:          $\mathcal{L}_{\text{Warm-Up}} = \mathbb{E}_{(a_i, v_i) \sim \mathcal{D}_l} H(\mathcal{G}_i, \mathcal{P}_i^t)$ {Supervised learning on labeled data.}
6:       **end while**
7:       Get the pseudo-labels $\mathcal{M}_i^{t,A}, \mathcal{M}_i^{t,B}$ from dual teachers
8:       **if** IoU$(\mathcal{M}_i^{t,A}, \mathcal{M}_i^{t,B}) \geq \tau$ **then**
9:          $\mathcal{IPL}(a_i, v_i) = \mathcal{M}_i^{t,A} \cdot \mathcal{M}_i^{t,B}$ {Compute Intersection of Pseudo-Labels (IPL).}
10:         $\hat{\mathcal{G}}_i = \mathcal{IPL}(a_i, v_i)$ {Update the pseudo-label $\hat{\mathcal{G}}_i$ of unlabeled data.}
11:         Add $(a_i, v_i)$ to new dataset $\mathcal{D}'_u$
12:       **end if**
13:    **end for**
14:    $\mathcal{D}_{mix} = \mathcal{D}_l \cup \mathcal{D}'_u$ {Mix the filtered unlabeled data and labeled data.}
15:    $\mathcal{L}_{\text{full}} = \left(\mathcal{L}_{\text{sup}}^A + \mathcal{L}_{\text{sup}}^B\right) + \lambda_u \left(\mathcal{L}_{\text{unsup}}^A + \mathcal{L}_{\text{unsup}}^B\right)$. {Students learning.}
16:    $\theta_m^t \leftarrow \beta \theta_{m-1}^t + (1 - \beta) \theta_m^s$ {Students update teachers via EMA.}
17: **end while**
18: **Return:** Dual teachers and students model parameters.

---

**NOTING TIPS:**

**Train.** Warm-Up Stage is essentially a supervised learning. The performance gains of subsequent Unbiased-Learning Stage over Warm-Up Stage is actually the performance gains of our semi-supervised framework over vanilla supervised training on the same labelled dataset $\mathcal{D}_l$, which proves the validity of the proposed Dual Mean-Teacher, as shown in the main results in Table 1 and Table 2.

**Inference.** For the localization result of $i_{th}$ audio-visual pair, we merge the outputs of the dual teachers to create a predicted map as below. Comparison of different predicted maps are described in C.2.

$$\mathcal{P}_i = \frac{1}{2}(\mathcal{P}_i^{t,A} + \mathcal{P}_i^{t,B}). \tag{15}$$

# B Experimental Settings

## B.1 Datasets

We have conducted our training and evaluation of the Progressing Teacher on two large-scale audio-visual datasets: Flickr-SoundNet and VGG-Sound, which consist of millions of unconstrained videos and $5,000$ and $5,158$ annotated samples, respectively. Each audio-visual pair is comprised of a single image frame from each video clip and an audio segment centered around it. The annotations are provided in the form of bounding boxes. The relevant information is presented in the Table 8.

Table 8: Datasets overview.

| | All Labeled Data | | | | | Test Set | | | | | Labeled Split | | | |
|---|---|---|---|---|---|---|---|---|---|---|---|---|---|---|
| | small | medium | large | huge | total | small | medium | large | huge | total | train | val | test | total |
| Flickr-SoundNet | 3 | 254 | 687 | 4056 | 5000 | 0 | 9 | 83 | 158 | 250 | 4250 | 500 | 250 | 5000 |
| VGG-SoundSource | 134 | 1796 | 1726 | 1502 | 5158 | 8 | 86 | 83 | 73 | 250 | 4250 | 500 | 250 | 5000 |

Furthermore, for the purpose of assessing the generalizability of our model, we have extended DMT to music domain (distribution), including: MUSIC-solo, MUSIC-duet, and MUSIC-Synthetic. The MUSIC dataset [37] comprises 685 untrimmed videos, encompassing 536 solo performances and 149 duet renditions, spanning across 11 distinct categories of musical instruments. The MUSIC-Synthetic [31, 52] is a multifaceted assemblage wherein four disparate solo audio-visual pairs of divergent classifications are randomly mixed, retaining solely two out of the four audio segments. This deliberate curation aligns aptly with the evaluation of discerningly sounding object localization.

## B.2 Backbones: VGGish and SoundNet

For audio backbones, we employ pre-trained VGGish and SoundNet. VGGish is pre-trained on AudioSet as audio feature extractors. The raw 3s audio signal is resampled at 16kHz and further transformed into $96 \times 64$ log-mel spectrograms as the audio input. The output is 128D vector. SoundNet takes the raw waveform of the 3s audio clip as input and produces a 1401D vector as output, which concatenates the 1000D object-level feature and the 401D scene-level feature, which are both obtained from different conv8 layer. Our main focus is to train the nonlinear audio feature transformation function, $g(\cdot)$, which is instantiated with two fully connected networks and a ReLU layer, to transform the network output feature into a 512D representation.

## B.3 Metrics: CIoU, MSE, F1 Score, Precision

We consider a set of audio-visual pairs as $\mathcal{D} = \{(v_i, a_i), \mathcal{G}_i\}$, where $\mathcal{G}_i$ is the ground-truth. We set $\mathcal{P}_i(\delta) = \{(x,y)|\mathcal{P}_i(x,y) > \delta\}$ is the foreground region of predicted map, and $\mathcal{G}_i(x,y) = \{(x,y)|\mathcal{G}_i(x,y) > 0\}$ is the foreground region of ground truth.

**CIoU.** The IoU of predicted map and ground truth can be calculated by:

$$IoU_i(\delta) = \frac{\sum_{x,y \in \mathcal{P}_i(\delta)} \mathcal{G}_i(x,y)}{\sum_{x,y \in \mathcal{P}_i(\delta)} \mathcal{G}_i(x,y) + \sum_{x,y \in \{\mathcal{P}_i(\delta) - \mathcal{G}_i\}} 1}. \tag{16}$$

In previous works, CIoU quantifies the proportion of samples with IoU value exceeding a predetermined threshold, typically set at 0.5.

**MSE.** MSE measures the difference between two maps on a pixel-wise basis, making it more suitable for evaluating dense prediction tasks than IoU. Other two metrics for small objects localization.

$$MSE_i = \frac{1}{HW} \sum_{x=1}^{W} \sum_{y=1}^{H} (\mathcal{P}_i(x,y) - \mathcal{G}_i(x,y))^2. \tag{17}$$

**Max-F1 and AP.** To compute true positives, false positives and false negatives, we closely follow SLAVC [10]. Then we can compute the precision and recall:

$$\text{Precision} = \frac{|\mathcal{TP}|}{|\mathcal{TP}| + |\mathcal{FP}|}, \qquad \text{Recall} = \frac{|\mathcal{TP}|}{|\mathcal{TP}| + |\mathcal{FN}|}. \qquad (18)$$

Then we compute F1 for all values of $\delta$ and report the Max-F1 score:

$$\text{F1} = \frac{2 * \text{Precision} * \text{Recall}}{\text{Precision} + \text{Recall}}, \qquad \text{max-F1} = \max(\text{F1}). \qquad (19)$$

Average Precision (AP) is the area under the precision-recall curve above. For a detailed calculation of max-F1 and AP, please refer to the SLAVC [10].

### B.4 Implementation details

In addition to the experimental settings mentioned in the main text, we used a batch size of 128. Warm-Up stage is trained for 6 epochs to achieve convergence, while the Unbiased-Learning stage is trained for 20 epochs. The learning rate for the image is set to 1e-4, and the weight for the contrastive loss $\lambda_u$ is set to 1. An Exponential Moving Average (EMA) decay of 0.999 is applied. The Adam optimizer is used for training, and the training is conducted on two GPUs. Our supplementary experiments were conducted on the Flickr-10k or Flickr-144k dataset, which contains 4k annotations. The trained models were evaluated on the Flickr-SoundNet testset.

### B.5 Baselines

- Attention 10k [14, 15] (CVPR$_{2018}$): introduce a dual-stream network and leverage an attention mechanism to capture the salient regions in semi-supervised or self-supervised environments.
- DMC [2] (CVPR$_{2019}$) : establish audio-visual clustering to associate sound centers with their corresponding visual sources.
- CoarsetoFine [32] (ECCV$_{2020}$) : leveraged a two-stage framework to capture cross-modal feature alignment between sound and vision.
- LVS [28] (CVPR$_{2021}$) : propose to mine hard negatives within an image-audio pair.

Table 9: Cross dataset performance. We train our model using the VGG-Sound 10k and 144k datasets and evaluate its performance on the Flickr-SoundNet dataset.

| Trainset | Methods | Flickr testset | |
|---|---|---|---|
| | | CIoU | AUC |
| VGG-Sound 10k | attention10k | 52.20 | 50.20 |
| | LVS | 61.80 | 53.60 |
| | EZVSL | 65.46 | 54.57 |
| | SLAVC | 74.00 | 57.74 |
| | SSPL | 76.30 | 59.10 |
| | SSL-TIE | 77.04 | 60.36 |
| | Ours($|\mathcal{D}_l| = 256$) | 85.04 (80.08) | 65.06 (60.14) |
| | Ours($|\mathcal{D}_l| = 2k$) | 87.36 (81.60) | 67.38 (61.26) |
| | Ours($|\mathcal{D}_l| = 4k$) | **88.20 (82.88)** | **67.56 (62.06)** |
| VGG-Sound 144k | attention10k | 66.00 | 55.80 |
| | LVS | 71.90 | 58.20 |
| | EZVSL | 79.51 | 61.17 |
| | SLAVC | 80.00 | 61.68 |
| | SSPL | 76.70 | 60.50 |
| | SSL-TIE | 79.50 | 61.20 |
| | Ours($|\mathcal{D}_l| = 256$) | 87.04 (80.08) | 64.72 (60.14) |
| | Ours($|\mathcal{D}_l| = 2k$) | 88.32 (81.60) | 67.78 (61.26) |
| | Ours($|\mathcal{D}_l| = 4k$) | **89.84 (82.88)** | **68.64 (62.06)** |

- EZVSL [9] (ECCV$_{2022}$) : introduce a multi-instance contrastive learning framework that utilizes Global Max Pooling (GMP) to focus only on the most aligned regions when matching audio and visual inputs.
- SLAVC [10] (NeurIPS$_{2022}$) : adopts momentum encoders and dropout to address overfitting and silence issues in single-source sound localization.
- SSPL [29] (CVPR$_{2022}$) : propose a negative-free method to extend a self-supervised learning framework to the audio-visual data domain for sound localization
- SSL-TIE [30] (ACM-MM$_{2022}$): introduce a self-supervised framework with a Siamese network with contrastive learning and geometrical consistency.

# C Comprehensive Experimental Results

## C.1 Cross-dataset Evaluation

To further validate the generalization ability of DMT, we conducted cross-dataset validation experiments. The results in Table 9 show that DMT still stays ahead, confirming the high generalization ability of our model.

## C.2 Different Predicted Map

In this section, we compare the accuracy of different predicted maps for sound localization. We evaluate individual predicted maps and a fused map as the final localization map, as defined by Eq. 15. Training is performed on the Flickr144k dataset using dual teacher results, as shown in Table 10. We find that fused predicted map from dual teachers with different backbones achieves better localization performance than from individual maps, which can be attributed to the fact that considering both localization results helps mitigate biases inherent in a single model.

Table 10: Results of different inference strategies.

|  | CIoU | AUC |
| --- | --- | --- |
| Student A | 86.20 | 66.16 |
| Student B | 86.80 | 66.84 |
| Fused Students | 88.60 | 68.56 |
| Teacher A | 87.20 | 67.57 |
| Teacher B | 87.60 | 67.98 |
| **Fused Teachers** | **90.40** | **69.36** |

Additionally, we assess the performance of teachers and students by comparing their fused predicted maps obtained during the same training session. The results, as shown in Table 10, indicate that **teachers outperform students**, which aligns with our expectations and further validates the effectiveness of our model.

## C.3 Effect of Warm-Up Stage

This section focuses on the analysis of convergence speed and the influence of the Warm-Up performance on the final results.

**Convergence Speed** Initially, we investigate the convergence speed of the Warm-Up stage with varying amounts of labeled data, as depicted in Figure 6. Notably, all supervised models exhibit rapid convergence within a specific number of epochs. Furthermore, as the quantity of data increased, the convergence speed decreases while simultaneously achieving higher levels of performance.

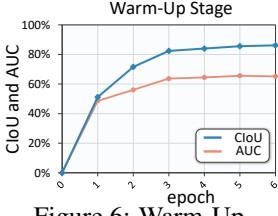
Figure 6: Warm-Up.

**Effect of Warm-Up Performance.** Subsequently, we investigate how the Warm-Up performance affects final results by experimenting with models that achieved different levels of convergence using the same amount of data. Training is performed on the Flickr144k dataset using dual teacher results, as presented in Table 11. The results indicate that better performance of Warm-Up stage leads to better final model performance, which can be attributed to higher-quality pseudo-labels and improved noise filtering, reducing confirmation bias. Conversely, the model exhibits the poorest performance in the absence of Warm-Up stage.

Table 11: Effect of Warm-Up Performance.

| Warm-Up | | Final | |
| --- | --- | --- | --- |
| CIoU | AUC | CIoU | AUC |
| 0 | 0 | 84.32 | 64.52 |
| 51.20 | 48.62 | 87.28 | 67.18 |
| 71.60 | 56.08 | 89.04 | 68.26 |
| **86.20** | **65.56** | **90.40** | **69.36** |

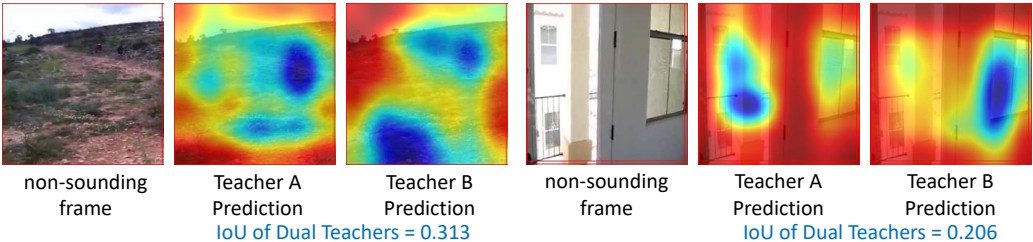

| non-sounding frame | Teacher A Prediction | Teacher B Prediction | non-sounding frame | Teacher A Prediction | Teacher B Prediction |

IoU of Dual Teachers = 0.313    IoU of Dual Teachers = 0.206

Figure 7: False-Positive Rejection Capability of Noise Filtering.

**Overall**, supervised audio-visual source localization demonstrates ease of convergence without requiring excessive training resources. Moreover, our proposed semi-supervised model consistently outperforms the supervised model by approximately $3\%$ in terms of absolute performance, validating its effectiveness.

## C.4 False-Positive Rejection Capability of Noise Filtering

After analyzing the filtered-out samples, we observed that the two independent teachers exhibit disagreement in localizing non-sounding objects. In such cases, the IoU falls significantly below the threshold, enabling the Dual Teachers to identify and reject non-sounding samples, which can be considered as false positives, as illustrated in Figure 7. Additionally, different filter thresholds represents different levels of filtering strictness, as detailed in Section C.6.

Furthermore, we analyzed the visual results of some noisy samples, as depicted in Figure 10. One can observe that frames without distinguishable sound objects or sounds that cannot be accurately represented by a bounding box (*e.g.*, wind sounds) can be easily identified through the inconsistency between the predictions of the two teachers.

## C.5 Hyper-parameters for Filter, IPL, and EMA

**Effect of Pseudo-Labeling Threshold.** The threshold $\delta$ is used to convert the predicted map into a binary map, as described in Eq.(6). In this section, we analyze the impact of different thresholds on pseudo-labels and the model. Training is conducted on the Flickr10k dataset. Figure 8 shows the results. A small delta value (*e.g.* $\delta = 0.5$) creates a large foreground area, introducing excessive noise and causing performance degradation as training progresses. On the other hand, A large value of $\delta$ (*e.g.* $\delta = 0.9$) indicates a small foreground area, causing the intersection between Dual Teachers to be minimal and resulting in samples being falsely rejected as noise, thus disturbing the model. Therefore, we choose $\delta = 0.6$ as the optimal threshold for our final selection.

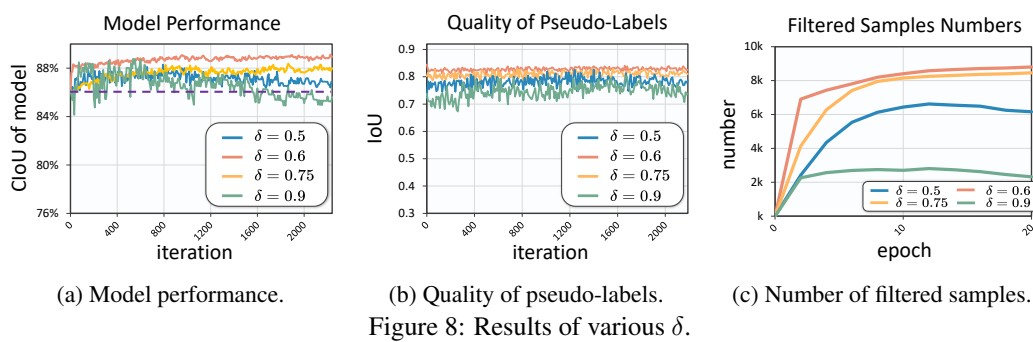

| (a) Model performance. | (b) Quality of pseudo-labels. | (c) Number of filtered samples. |

Figure 8: Results of various $\delta$.

**Effect of Filtering Threshold.** In Section 4.2, we employ a confidence threshold, denoted as $\tau$, to filter out noisy samples, which are more likely to be false-positive instances. We evaluate the effect of different threshold values $\tau$. As shown in Figure 9, As the threshold value $\tau$ increases from 0 to 0.9, the number of accepted samples decreases. However, setting a very high threshold (e.g., $\tau = 0.9$) leads to unsatisfactory results due to the limited number of accepted samples, reducing the available information from unlabeled data. Conversely, using a low threshold (e.g., $\tau = 0.6$) introduces a confirmation bias from noisy samples, hindering favorable outcomes. Upon analysis, we discover that the performance shows little variation between threshold values of $\tau = 0.7$ and $\tau = 0.8$, indicating

a balance between unlabeled information and bias within the 0.7-0.8 range. As a result, we opt for $\tau = 0.7$ as the preferred threshold for our final selection.

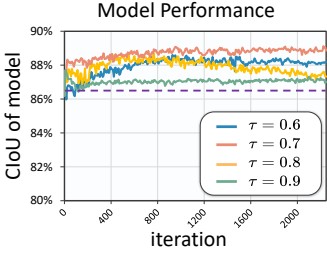
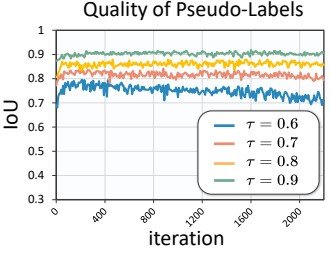
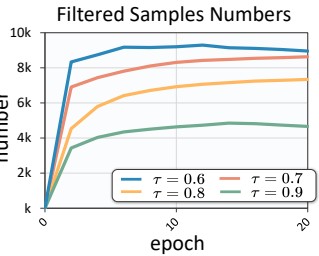

(a) Model performance.  (b) Quality of pseudo-labels.  (c) Number of filtered samples.

Figure 9: Results of various $\tau$.

**Effect of EMA Rates** We also examine the model performance with various exponential moving average (EMA) decay values, denoted as $\beta$, ranging from 0.9 to 0.999, and present the results of the teachers in Table 12. We observe that a smaller EMA decay leads to a faster update rate, lower CIoU, and higher variance. Conversely, a larger EMA decay value results in slower learning for the teachers. Therefore, we select an appropriate EMA decay value of $\beta = 0.999$ to strike a balance between the update rate and the stability of the learning process.

Table 12: Results on various EMA $\beta$.

|  | $\beta$ | CIoU | AUC |
|---|---|---|---|
| Flickr 10k | 0.9 | 86.48 | 65.16 |
|  | 0.99 | 88.64 | 66.94 |
|  | **0.999** | **88.80** | **67.81** |
| Flickr 144k | 0.9 | 87.84 | 85.82 |
|  | 0.99 | 89.92 | 68.86 |
|  | **0.999** | **90.40** | **69.36** |

## C.6 Effect of Data Augmentation

We evaluate the effect of RandAug [51] on a supervised model on 4k labeled data, as shown in Table 13. Without data augmentation, the model exhibits significant over-fitting. With RandAug, this issue is mitigated, which indicates that RandAug serves not only as a means of consistency regularization but also as a method to enhance the model's generalization performance.

Table 13: Results of data augmentation (*i.e.*, RandAug.).

|  | Trainset | | Testset | |
|---|---|---|---|---|
|  | CIoU | AUC | CIoU | AUC |
| w/o RandAugment | 88.20 | 67.82 | 84.80 | 60.44 |
| w/ RandAugment | 87.68 | 67.54 | 86.20 | 65.56 |

## C.7 IPL on Different Object Size

We assess the adaptability of IPL to various object sizes, and compare with existing methods, two teachers with DMT. Table 14 results highlight prior methods' diminishing performance with smaller objects, while DMT consistently excels across all size subsets. This enhancement is attributed to Filtering and IPL synergy. Under the filtering mechanism, only highly similar pseudo-labels can contribute to model training. This keeps the intersection of pseudo-labels consistently aligned with object sizes. If pseudo-labels decrease significantly, IoU declines, excluding noisy samples from training. Moreover, in the second-stage training, we use labeled data to prevent size bias and ensure unbiased treatment of objects of all sizes.

Table 14: Performance across various sizes of sounding objects.

| Size | SLAVC | | teacher1 | | teacher2 | | DMT | |
|---|---|---|---|---|---|---|---|---|
|  | MSE ↓ | IoU ↑ | MSE ↓ | IoU ↑ | MSE ↓ | IoU ↑ | MSE ↓ | IoU ↑ |
| small | 0.705 | 2.10 | 0.213 | 2.58 | 0.183 | 2.26 | **0.205** | **2.65** |
| medium | 0.235 | 22.00 | 0.156 | 12.47 | 0.176 | 12.28 | **0.164** | **33.50** |
| large | 0.427 | 48.11 | 0.202 | 55.32 | 0.221 | 54.68 | **0.212** | **55.50** |
| huge | 0.358 | 61.64 | 0.212 | 66.84 | 0.217 | 66.26 | **0.215** | **67.70** |

## C.8 How to avoid model collapse?

There are diversity and individuality between two teachers, as in Q2, which helps to prevent two teachers convergence to one model. The noisy filter module of DMT selects 'stable samples' via consensus and assigns high-quality pseudo-labels with IPL, such spirit has been validated by prior work that 'stable samples' could help avoid model collapse. Two teachers are first trained in Warm-Up stage for better initialization. Moreover, in stage-2, we also include supervised training on labeled data and contrastive learning on unlabeled data, the two objectives would ensure the model possesses robust localization capabilities over the course of stage-2. The results in Table 15 validate each component to avoid model collapse.

Table 15: Model collapse results. $\mathcal{A}$, $\mathcal{B}$ denotes augmentation and backbone.

| method | DMT | same $\mathcal{A}$ | same $\mathcal{B}$ | w/o annotation in stage-2 | same $\mathcal{A}$ & $\mathcal{B}$ w/o annotation |
|---|---|---|---|---|---|
| CIoU | **90.4** | 87.2 | 85.4 | 81.6 | 7.2 |

## C.9 Quality Analysis

We present the visual localization results of DMT in Figure 10. It effectively locates objects of different sizes, distinguishes them from the background by clear boundaries, and demonstrates some multi-object localization capability. Notably, DMT learns semantic information and can precisely localize specific sound-producing regions instead of the entire object. For example, in the third row of the Figure 10 on the right, it accurately locates the mouth of a person rather than the entire person.

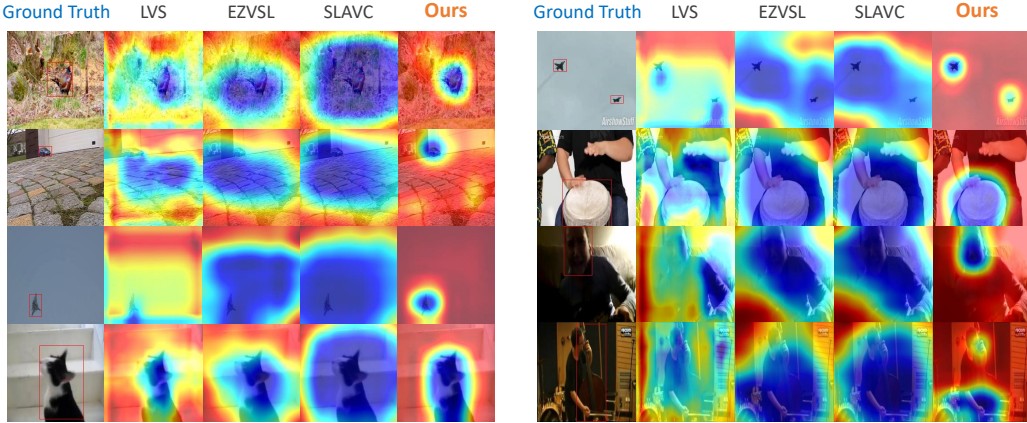

Figure 10: Visualizations of various methods.

## D   Further Analysis: Limitations in Existing AVSL and DMT

Based on the formula of contrastive loss, we can observe that the core idea of existing contrastive learning methods is to match the visual frames and corresponding audio clips within the same video as a whole. The audio-visual pairs from the same video are considered positive pairs, while the frames and audio clips from different videos are considered negative pairs. The contrastive loss aims to maximize the similarity between positive samples and minimize the similarity between negative samples. The differences among existing self-supervised methods lie in the selection of the similarity function $s(\cdot)$ and the positive-negative sample pairs.

$$\mathcal{L}_{\text{unsup}} = -\mathbb{E}_{(a_i,v_i)\sim\mathcal{D}_u} \left[ \log \frac{\exp(s(g(a_i), f(v_i))/\tau_t)}{\sum_{j=1}^{n} \exp\left(s\left(g(a_i), f(v_j)\right)/\tau_t\right)} + \log \frac{\exp(s(f(v_i), g(a_i))/\tau_t)}{\sum_{j=1}^{n} \exp\left(s\left(f(v_i), g(a_j)\right)/\tau_t\right)} \right].$$

## D.1 Global and Local Information

In the given formula, different methods employ different match functions $s(\cdot)$ to compute the distance or similarity between positive samples. For instance, Attention10k [14, 15] uses the Euclidean distance, LVS [28] utilizes the Frobenius inner product, and EZVSL [9] applies Global Max Pooling:

$$
\begin{aligned}
\text{Attention10k:} \quad s(\cdot) &= \| f_{att}(v_i) - g(a_i) \|_2 \,, \\
\text{LVS:} \quad s(\cdot) &= \frac{1}{|\hat{m}_{ip}|} \langle \hat{m}_{ip}, \text{sim}\left(f(v_i), g(a_i)\right) \rangle \,, \\
\text{EZVSL:} \quad s(\cdot) &= \max \text{sim}\left(f(v_i), g(a_i)\right) \,, \\
\text{SLAVC:} \quad s(\cdot) &= \sum_{x,y} \rho\left(\frac{1}{\tau} \text{sim}\left(g^{\text{loc}}(a_i), f^{\text{loc}}(v_i)\right)\right) \cdot \rho\left(\frac{1}{\tau} \text{sim}\left(g^{\text{avc}}(a_i), f^{\text{avc}}(v_i)\right)\right) \,.
\end{aligned}
$$

All of these functions capture the overall matching degree between audio and global visual representations. However, after the computation of $s(\cdot)$, the model loses the positional information of the two-dimensional visual representation. This positional information is crucial for fine-grained localization tasks.

## D.2 Position-Aware Contrastive Loss

We refer to the methods that incorporate position information as 'position-aware'. In the above formulas, we can observe that the distances or similarities between samples are calculated in a position-aware manner. For example, in the Attention10k [14, 15] method, the attention mechanism $f_{att}$ takes into account the positional information. Similarly, in LVS [28], the foreground mask $\hat{m}_{ip}$ distinguishes the background as hard negatives, incorporating the positional context. EZVSL [9] uses the maximum value to capture the positional information, while SLAVC [10] incorporates localization information. Taking LVS [28] as an example, it specifically treats the background of the image as hard negatives, effectively leveraging the positional cues for discrimination and learning.

$$
P_i = \frac{1}{|\hat{m}_{ip}|} \langle \hat{m}_{ip}, \text{sim}(g(a_i), f(v_i)) \rangle \,,
$$

$$
N_i = \frac{1}{|\mathbf{1} - \hat{m}_{in}|} \langle \mathbf{1} - \hat{m}_{in}, \text{sim}(g(a_i), f(v_i)) \rangle + \frac{1}{hw} \sum_{j \neq i} \langle \mathbf{1}, \text{sim}(g(a_i), f(v_j)) \rangle \,,
$$

$$
\mathcal{L}_{unsup} = -\frac{1}{k} \sum_{i=1}^{k} \left[ \log \frac{\exp(P_i)}{\exp(P_i) + \exp(N_i)} \right] \,.
$$

where, $\hat{m}_{ip}$ is the mask of foreground, which strongly relies on the initialization of the model. According to the formula, both the positive $(P_i)$ and negative $(N_i)$ samples in the training process are influenced by the initial values of the foreground mask $\hat{m}_{ip}$. This implies that the model's localization results are heavily dependent on the initialization.

## D.3 Initialization

The different matching mechanisms, represented by the function $s(\cdot)$, rely on the initialization of the entire visual model, specifically the pre-trained ResNet-18 [47, 48], where the average of the pixel-wise features is taken as the initial result at epoch 0. This initialization result serves as the basis for the computation of position-aware components, such as the attention mechanism or Global Max Pooling (GMP). Subsequently, during the model's training, these initial localization results are reinforced and refined. However, if the initial localization results are inaccurate (which is often the case), subsequent training may have difficulty detecting and correcting these inaccuracies. As a result, the errors may accumulate over time without being effectively addressed, leading to degraded performance.

## D.4 False Positives, False Negetives and Multi-Source

From the contrastive learning formula, it is apparent that contrastive learning assumes the presence of sound-producing objects in the visual input and enforces alignment between highly confident

visual regions and their corresponding audio features. However, pure contrastive learning, without the incorporation of additional modules, cannot directly reject non-sounding samples. Recently, some works have recognized this limitation and started to investigate the presence of sound-producing objects in images and tackle the task of multi-source sound localization. Examples of such works include DSOL [31], IER [34], and AVGN [11].

Furthermore, due to the absence of class labels during the selection of positive and negative samples, visual-audio pairs belonging to the same sound-producing object class but originating from different videos are still treated as negative samples, resulting in a false negatives issue. Several methods have emerged to address this problem, as highlighted in [35, 36].

In addition, the commonly used matching mechanism, Global Max Pooling, is suitable only for single-source localization since it focuses solely on the region with the highest confidence, neglecting other potential sound-producing objects.

These three aforementioned challenges cannot be effectively resolved solely through simple models or algorithms without positional annotations. Therefore, they have become prominent research areas that are currently receiving considerable attention.

## D.5 Limitations of DMT

DMT does not involve class information, so it struggles to localize among fine-grained objects due to poor discriminative ability. By incorporating category signals, models could better implement fine localization. Besides, DMT could not handle multi-object localization well. We will devise specialized components to address this issue.

