# OpenReview forum: "Dual Mean-Teacher: An Unbiased Semi-Supervised Framework for Audio-Visual Source Localization"
_NeurIPS.cc/2023/Conference — NeurIPS 2023 poster_

### Official Review · Reviewer_1pQt · 2023-07-03

**Soundness:** 3 good
**Presentation:** 3 good
**Contribution:** 3 good
**Rating:** 6
**Confidence:** 4

**Summary:**

The authors propose a two teacher-student architectures, pre-trained with limited labeled data in the semi-supervised learning framework to solve audio-visual source localization. They show that proposed approach can reach state-of-the-art performance in Flickr-SoundNet and VGG-SS. The authors also provide ablation studies and analysis in the characteristics of the dataset utilized.

**Strengths:**

- The proposed approach to utilize semi-supervised learning techniques on audio-visual source localization is an interesting and original idea.
- The authors provide thorough ablation studies on the choices of semi-supervised learning configurations, as well as backbones selections.
- The analysis of data involved in this paper provides new insights and different perspective to the problem, including the analysis of small objects and false positives.

**Weaknesses:**

- Consider extend to more audio-visual source localization datasets such as MUSIC-synthetic, MUSIC-duet and AudioSet-instrument-multi described in the DSOL paper. This can help provide more insights on the proposed system in diverse of datasets including different audio domain such as music, and different distribution of sounding objects, number of bounding boxes in the image.
- For the subsets mentioned in Section 5.2, it would be helpful to provide more definition on how small objects are defined, how false positives are measured, and the definition of open set.
- Minor comments: in Section 5.2 line 211, consider explicitly refer those numbers to Table 1.

**Questions:**

- In Section 4.2 the proposed intersection of pseudo-labels (IPL) already bias toward small regions. How do you make sure that there is no harm to large regions for proposed approach?
- In Table 4, what is the datasets utilized for this comparisons?

**Limitations:**

This paper can benefit from applying to more datasets involving different audio domains and distributions of annotations.

---

> ### Author Rebuttal · Authors · 2023-08-08
>
> Thank you for your insightful advice and valuable questions, we will respond to your concerns point by point.
>
> > Q1: Consider extend to more audio-visual source localization datasets.
> * We have supplemented the evaluation on the music domain. Due to the absence of class information, we did not compare our approach with DSOL and its evaluation metrics.
> * Firstly, we directly evaluated the models trained with VGG-ss 144k on MUSIC-solo, MUSIC-duet MUSIC synthetic, and Audioset-multi, The setting and data splits follow DSOL [A]. We compare DMT with prior SLAVC, and SLAVC with naive semi-supervised learning (SSL), supervised learning, where naive SSL means models are trained on labeled and unlabeled data, with $\mathcal{L}_\text{sup}$ and  $\mathcal{L}_\text{unsup}$ respectively. Results (CIoU) are shown as follows.
>
> |     methods     | solo  | duet  | synthetic | audioset-multi |
> | :-------------: | :---: | :---: | :-------: | :------------: |
> |      SLAVC      | 73.28 | 6.00  |   3.86    |     26.74      |
> | SLAVC+naive SSL | 67.23 | 12.86 |   7.62    |     29.82      |
> | supervised DMT  | 75.30 | 9.66  |   4.17    |     28.16      |
> |       DMT       | 89.12 | 22.14 |   26.59   |     36.68      |
>
>
> * As shown above, DMT could generalize well in the music domain but with degraded results in multiple objects scenarios. From Figure 3 in rebuttal PDF, one could observe that DMT is capable of accurately localizing instruments in the image based, however, it lacks the ability to discern which instrument is producing sound and which one is silent based solely on the audio input.
> * We also finetune DMT on MUSIC-solo dataset, results in PDF show that DMT could gain partial improvements and is capable of finer-grained discrimination among certain instruments. However, due to the absence of class information, there is still significant room for improvement with multiple sounding objects. We plan to address this issue in future work.
>
>
> > Q2: Provide more definition of small objects, false positives, and open set.
>
> Thank you for your advice. We will incorporate definitions for small objects, false positives, and open set. The revised part is as follows:
> * **small objects:** We categorize objects based on their bounding box pixel area. Small: $1\sim 32^2$, medium: $32^2\sim 96^2$, large: $96^2\sim 144^2$, and huge: $144^2\sim 224^2$.
> * **false positives:** Sounding objects are sometimes not visible in the image (off-screen). Such incongruous audio-visual pairs are often treated as positive samples in audio-visual contrastive learning, as they originate from the same video. Consequently, they become false positives.
> * **Open set:** It is important to investigae whrther model could generalize to unseen categories [B]. We randomly sample 110 categories (heard) subset from VGG-ss for training, and test our model on the other 110 categories (unheard) subset. The categories of the two subsets are disjoint.
>
>
> > Q3: Minor comments: in Section 5.2 line 211, consider explicitly refer to Table 1.
>
> Thanks. We will refer Table 1 explicitly.
>
>
> > Q4: In Section 4.2 the proposed IPL already bias toward small regions. How to make sure there is no harm to large regions?
>
> * Firstly, IPL captures the intersection of pseudo-labels generated independently by two teachers. It employs the consensus of both models to rectify the pseudo-labels, yielding more accurate and high-quality labels, rather than progressively diminishing the intersection in order to bias towards small objects.
> * Secondly, both large and small objects have their appropriately sized pseudo-labels. Under the filtering mechanism, only highly similar pseudo-labels can contribute to model training. In this context, the intersection of pseudo-labels stabilizes consistently, maintaining proximity to the size of the respective objects. If the pseudo-label drastically reduces, the IoU decreases, preventing noisy samples from participating in model training.
> * Lastly, during the second-stage training, we continue to employ labeled data for supervised learning, which ensures the avoidance of bias towards small objects.
> * We have conducted experiments of various sizes. DMT shows significant improvements across all subset sizes. IPL in DMT is stable and non-detrimental. Results are shown below.
>
> |        | SLAVC |       | teacher1 |       | teacher2 |       | DMT  |           |
> | :----: | :---: | :---: | :------: | :---: | :------: | :---: | :--------: | :-------: |
> |        | MSE↓  | IoU↑  |   MSE↓   | IoU↑  |   MSE↓   | IoU↑  |    MSE↓    |   IoU↑    |
> | small  | 0.705 | 2.10  |  0.213   | 2.58  |  0.183   | 2.26  | **0.205**  | **2.65**  |
> | medium | 0.235 | 22.00 |  0.156   | 12.47 |  0.176   | 12.28 | **0.164**  | **33.50** |
> | large  | 0.427 | 48.11 |  0.202   | 55.32 |  0.221   | 54.68 | **0.212**  | **55.50** |
> |  huge  | 0.358 | 61.64 |  0.212   | 66.84 |  0.217   | 66.26 | **0.215**  | **67.70** |
>
>
> > Q5: What is the datasets utilized in Tab. 4?
>
> Here, we train and test on Flickr-10k.
>
>
> > Q6: This paper can benefit from applying to more datasets involving different audio domains and distributions of annotations.
>
> Thank you for your advice. We have included experimental results across different domains, varying numbers of sound sources, and realistic datasets, along with appropriate citations to relevant papers. We have enhanced our analysis of the framework. For future work, we will focus on dataset results across various domains.

---

> > ### Comment · Reviewer_1pQt · 2023-08-18
> >
> > Thank the authors for your thorough responses and extra experimental results addressing the questions. Could you please include the experimental results and definitions in the paper or the appendix for the final version, this helps to provide a more complete view of your work. I am therefore changing the rating.

---

> > > ### Author Response · Authors · 2023-08-20
> > > **Thanks for Acknowledgement and Present the Revisions**
> > >
> > > Thank you for your comment! We are greatly encouraged by your acknowledgment.
> > >
> > > Certainly, we are more than willing to include the experimental results and definitions in the final version of the paper or in the appendix. This will enhance the comprehensiveness of our article and provide deeper insights into our method. We have incorporated the content of this rebuttal into our paper's revised version, which is as follows:
> > >
> > > **Datasets (Ln185-189)**
> > > > Furthermore, for the purpose of assessing the generalizability of our model, we have extended DMT to music domain (distribution), including: MUSIC-solo, MUSIC-duet, MUSIC-Synthetic, and AudioSet-instrument. The MUSIC dataset[1] comprises 685 untrimmed videos, encompassing 536 solo performances and 149 duet renditions, spanning across 11 distinct categories of musical instruments. The MUSIC-Synthetic[2,3] is a multifaceted assemblage wherein four disparate solo audio-visual pairs of divergent classifications are randomly mixed, retaining solely two out of the four audio segments. This deliberate curation aligns aptly with the evaluation of discerningly sounding object localization. The AudioSet-instrument dataset[4] forms a subset within the broader AudioSet framework. More details are shown in appendix.
> > >
> > >
> > > **Experiments**
> > >
> > > Ln214
> > >
> > > > Specifically, we randomly sample 110 categories (heard) subset from VGG-ss for training, and test on the other 110 categories (unheard) subset, in order to evaluate model in open world. The categories of the two subsets are disjoint.
> > >
> > > Ln216
> > >
> > > > **Capacity for Cross-Domain Generalization and Multi-Source Localization.**
> > > >
> > > > We tested DMT's generalization across different domains and its ability to localize multiple objects. Models trained using VGG-ss 144k were directly evaluated on MUSIC-solo, MUSIC-duet, MUSIC synthetic, and Audioset-multi datasets. Results in Table 4 demonstrate DMT's strong generalization in the music domain, outperforming other methods and the supervised model. As shown in Figure 10 (Appendix), previous methods struggle to accurately localize multiple sounding objects, either missing them or including all sounding objects within a large foreground area. In contrast, DMT localizes each instrument accurately and seperately. However, without category information for fine-grained training, it leads to suboptimal performance to differentiate between multiple active and silent instruments. We plan to address this issue in future work.
> > >
> > > Ln217
> > >
> > > > We categorize objects based on their bounding box pixel area into small ($1\sim 32^2$), medium ($32^2\sim 96^2$), large ($96^2\sim 144^2$), and huge ($144^2\sim 224^2$).
> > >
> > > Ln232
> > >
> > > > Furthermore, we assessed the model's capability to identify false positives, signifying instances where sounding objects are occasionally not visually observable within the image (off-screen).
> > >
> > > **Appendix**
> > >
> > > C.7 IPL on Different Object Size
> > >
> > > > We assess the adaptability of IPL to various object sizes, and compare with existing methods, two teachers with DMT. Table 13 results highlight prior methods' diminishing performance with smaller objects, while DMT consistently excels across all size subsets. This enhancement is attributed to Filtering and IPL synergy.
> > > Under the filtering mechanism, only highly similar pseudo-labels can contribute to model training. This keeps the intersection of pseudo-labels consistently aligned with object sizes. If pseudo-labels decrease significantly, IoU declines, excluding noisy samples from training. Moreover, in the second-stage training, we use labeled data to prevent size bias and ensure unbiased treatment of objects of all sizes.
> > >
> > >
> > >
> > > We hope the reviewer could acknowledge these revisions. And if you have further suggestion, we are more than willing to discuss with you.
> > >
> > > Thanks for your consideration!
> > >
> > >
> > > **Referrence:**
> > >
> > > [1] Hang Zhao, Chuang Gan, Andrew Rouditchenko, Carl Vondrick, Josh McDermott, and Antonio Torralba. The sound of pixels. In The European Conference on Computer Vision (ECCV), September 2018.
> > >
> > > [2] Di Hu, Rui Qian, Minyue Jiang, Xiao Tan, Shilei Wen, Errui Ding, Weiyao Lin, and Dejing Dou. Discriminative sounding objects localization via self-supervised audiovisual matching. Advances in Neural Information Processing Systems, 33:10077–10087, 2020.
> > >
> > > [3] Di Hu, Yake Wei, Rui Qian, Weiyao Lin, Ruihua Song, Jirong Wen. Class-aware sounding objects localization via audiovisual correspondence. IEEE Transactions on Pattern Analysis and Machine Intelligence, 44(12), 9844-9859, 2021.
> > >
> > > [4] J. F. Gemmeke, D. P. W. Ellis, D. Freedman, A. Jansen, W. Lawrence, R. C. Moore, M. Plakal, and M. Ritter. Audio set: An ontology and human-labeled dataset for audio events. In 2017 IEEE International Conference on Acoustics, Speech and Signal Processing (ICASSP), pages 776–780, March 2017.

---

### Official Review · Reviewer_MLW4 · 2023-07-05

**Soundness:** 4 excellent
**Presentation:** 3 good
**Contribution:** 3 good
**Rating:** 7
**Confidence:** 4

**Summary:**

This paper proposes a semi-supervised approach for audio-visual source localization on images based on an iterative student-teacher refinement scheme where the masks for the sounding objects are becoming gradually better by combining the estimated masks of two independent teacher models. The experimental results show that combining the estimates of the two teachers in such a way leads the student models to learn robust representations for the sounding objects under multiple datasets and yielding state-of-the-art results for the task of on-screen sound localization. The authors also provide experiments to show several ablation studies that show the importance of their design choices and the robustness of their proposed algorithm in different scenarios.


**Strengths:**

- The paper is clearly written and the ideas presented are detailed enough to be followed by researchers in the field for reproducibility.
- The experiments are extensive in multiple terms (ablation studies showing the importance of each design choice and each module, multiple datasets and metrics, including false positives and also analyzing the robustness of the algorithm).
- The experiments show really strong results for the proposed semi-supervised approach for on-screen sound localization and show great potential for becoming a benchmark method in the field.


**Weaknesses:**

The paper has a great potential to be a valuable contribution to the field of audio-visual on-screen sound localization but it needs to address the following concerns / issues to uncover its full potential.

- Using the word “unsupervised” is incorrect. The authors use this word to describe previous methods which all of them (to the best of my knowledge) still use pre-trained video/image encoders, pre-trained on the whole ImageNet and even sometimes some segmentation datasets. Thus, the authors should clearly distinguish all these works from what “unsupervised” truly means and properly define them as weakly or partially supervised on-screen sound localization approaches. This would not take away points from the novelty of this work . But in the current state, the manuscript’s multiple usages of the “unsupervised” word decrease its validity.
- How do the authors enforce that the two teachers are going to be producing different sets of predictions? Is this simply the different teacher update protocols being used for the teachers? To the best of my understanding this training method could collapse if the teachers agree in terms of their predictions and converge both to a noisy pseudo-mask. This can be extremely cumbersome when one moves away from more curated datasets to more real world datasets with possible unseen sounding objects / actions.
- Building upon my previous argument, using more realistic and large scale datasets would also show whether the proposed approach is robust when trained with videos that contain no on-screen objects which is very difficult to solve for related problems like on-screen sound separation [A, B]. I don’t want to extend the problem to cases where more than one sound appear on-screen / off-screen and thus some audio separation front-end might be needed before obtaining the audio features. But I would still expect some more experiments or discussion on how the authors could avoid potential problems like off-screen only videos with their method.

I would be more than happy to increase my score if the authors address all my concerns and answer my questions.

[A] Tzinis E, Wisdom S, Jansen A, Hershey S, Remez T, Ellis D, Hershey JR. Into the Wild with AudioScope: Unsupervised Audio-Visual Separation of On-Screen Sounds. In International Conference on Learning Representations 2021.

[B] Tzinis, E., Wisdom, S., Remez, T. and Hershey, J.R., 2022, October. Audioscopev2: Audio-visual attention architectures for calibrated open-domain on-screen sound separation. In European Conference on Computer Vision (pp. 368-385). Cham: Springer Nature Switzerland.


**Questions:**

How would the authors approach the problem of wanting to add more sound objects / actions that have not been seen during the pre-training stage of the first pseudo-mask estimators?


**Limitations:**

The authors discuss some limitations of their work and I think they should include the issues that I raised in the Weaknesses Section above and that they will not be able to address through experiments and/or rebuttal.

---

> ### Author Rebuttal · Authors · 2023-08-08
>
> Thank you for your insightful advice and valuable questions, we will respond to your concerns point by point.
>
> > Q1: Using the word “unsupervised” is incorrect.
>
> Thank you for your advice. We will revise '**unsupervised**' to '**weakly-supervised**' due to the usage of pre-trained models and clarify the difference from truly 'unsupervised' learning. Besides, following your suggestion, we will emphasize that the task mainly concerns on-screen sound localization and supplement discussion regarding off-screen scenarios in our paper.
>
>
> > Q2: How to ensure two teachers will produce different sets of predictions? Is this simply the different teacher update protocols?
>
> Dual teachers employ different backbones incorporated with distinct prior knowledge. Additionally, in stage-2, two students are fed the same image with two different strong data augmentations (pseudo-labels also require corresponding variations based on position augmentation), and in turn, they update two teachers via EMA. The aforementioned two points ensure the diversity between the two teachers.
>
>
> > Q3: How to avoid model collapse?
>
> * There are diversity and individuality between two teachers, as in Q2, which helps to prevent two teachers convergence to one model, which is validated in co-rectify paper [1].
> * The noisy filter module of DMT selects 'stable samples' via consensus and assigns high-quality pseudo-labels with IPL, such spirit has been validated by prior work [2] that 'stable samples' could help avoid model collapse.
> * Two teachers are first trained in WarmUp stage for better initialization. Moreover, in stage-2, we also include supervised training on labeled data and contrastive learning on unlabeled data, the two objectives would ensure the model possesses robust localization capabilities over the course of stage-2.
> * Results below validates each component for avoiding model collapse.
>
> | method | original DMT | same $\mathcal{A}$ | same $\mathcal{B}$ | w/o annotation in stage-2 | same $\mathcal{A}$ & $\mathcal{B}$ & w/o annotation |
> | :----: | :----------: | :----------------: | :----------------: | :-----------------------: | :-------------------------------------------------: |
> |  CIoU  |   **90.4**   |        87.2        |        85.4        |           81.6            |                         7.2                         |
>
> where $\mathcal{A}$ and $\mathcal{B}$ denotes augmentation and backbone.
>
>
> > Q4: More real world datasets with possible unseen sounding objects.
>
> * We have validated the effectiveness of DMT on real-world datasets, and in fact, VGG-ss is an in-the-wild dataset, encompassing various scenes and categories. As in Table 2, DMT achieves SOTA performance. However, there is indeed substantial room for improvement on real datasets, which is left for future work.
> * For the unseen objects localization issue, we have investigated open-set or cross-dataset generalization performance, as in Section 5.2, Table 3(c), we compare DMT with prior methods in terms of localization of unseen objects in the training-stage, DMT still achieves SOTA (CIoU 43.12 and AUC 42.81). Additionally, in Table 2 in Appendix, we have conducted cross-dataset experiments on unseen Flickr testset.
> * We also summarize the results here and add MUSIC reslts, as follows.
>
> |                   | openset |       | cross dataset |       | MUSIC |       |
> | :---------------: | :-----: | :---: | :-----------: | :---: | :---: | :---: |
> |      methods      |  CIoU   |  AUC  |     CIoU      |  AUC  | CIoU  |  AUC  |
> |    supervised     |  37.34  | 39.81 |     82.40     | 62.18 | 75.30 | 58.42 |
> | DMT w/o pre-train |  36.72  | 38.66 |     81.20     | 61.56 | 69.46 | 54.65 |
> | DMT w/ pre-train  |  43.12  | 42.81 |     89.84     | 68.64 | 82.12 | 61.86 |
>
>
>
> > Q5: Using more realistic and large scale datasets for off-screen scenarios.
>
> * DMT could leverage the consensus of two teachers to filter out off-screen samples, the results are shown in PDF file.
> * Off-screen is an important issue, we will add and cite papers [A,B] and take discussions in the revised version.
>
>
> > Q6: How to approach the problem with more sound objects / actions unseen during the pre-training stage of the first pseudo-mask estimators?
>
> * The pre-training gives backbones substantial prior knowledge, as it encounters numerous categories during the pretraining phase. The localization of unseen objects results degrade a lot if the backbone is trained from scratch, results are shown in **response to Q4**.
> * Models could leverage the similarity among objects to localize unseen but samples with similar semantics.
>
>
> > Q7: Include the raised issues unable to address through experiments.
>
> Thanks for your advice. The limitations are shown in the **General Response** part and we will incorporate them in the future version.
>
>
> **References:**
> [1]. Zhou Q, Yu C, Wang Z, et al. Instant-teaching: An end-to-end semi-supervised object detection framework. CVPR, 2021.
> [2]. Ke Z, Wang D, Yan Q, et al. Dual student: Breaking the limits of the teacher in semi-supervised learning. ICCV, 2019.

---

> > ### Comment · Reviewer_MLW4 · 2023-08-14
> > **Response**
> >
> > Thanks for addressing all my concerns! In my turn, as promised, I will also raise my score to a clear accept.

---

### Official Review · Reviewer_dVkQ · 2023-07-06

**Soundness:** 3 good
**Presentation:** 3 good
**Contribution:** 2 fair
**Rating:** 5
**Confidence:** 4

**Summary:**

The authors propose a semi-supervised learning method for audio-visual source localization, which applies consensus-based confidence estimation to filter out noisy samples. The algorithm maintains two independent teacher models for the pseudo labeling of unlabeled data and for the consensus evaluation, where they use exponential mean average to update the two teacher models from two student models. Experiments using Flicker-SoundNet and VGG Sound source significantly improve performance from existing methods.

**Strengths:**

The proposed method significantly improves the audio-visual source localization performance.

**Weaknesses:**

While empirical improvement is significant, the technical novelty is limited. Obtaining performance improvement by model aggregation is a popular concept.

**Questions:**

In equation (1), how do you use the global max-pooing and average-pooling to pair f and g?

**Limitations:**

Do you use the same (augmented) input and network structure for the two student models?
The two models should have individuality and perform similarly to make the consensus meaningful.
Are they optimal in this aspect, or is there room for improvement?

---

> ### Author Rebuttal · Authors · 2023-08-09
>
> Thank you for your insightful advice and valuable questions, we will respond to your concerns point by point.
>
>
> > Q1: While empirical improvement is significant, the technical novelty is limited. Obtaining performance improvement by model aggregation is a popular concept.
>
> * DMT is different from simple aggregation/ensembling. The primary objective of introducing dual teachers is to establish a filtering mechanism through consensus to mitigates confirmation bias. Dual teachers interact with each other and collectively filter select stable samples and assign high-quality pseudo-labels, as a result, the resulting models of both teachers are influenced by the other one.
> * To conclude, dual teachers in DMT interact with each other during training rather than simply ensembling the predictions.
> * Results below show that DMT outperforms naive ensembling of three models although DMT has only two models.
>
> |         method          | model 1 CIoU | model 2 CIoU | model 3 CIoU | ensembling inference CIoU |
> | :---------------------: | :----------: | :----------: | :----------: | :-----------------------: |
> | three models ensembling |     84.4     |     88.0     |     86.8     |           87.6            |
> |           DMT           |     89.2     |     88.8     |      -       |         **90.4**          |
>
>
>
> > Q2: In equation (1), how do you use the global max-pooing and average-pooling to pair f and g?
>
> I appreciate your highlighting of this concern. We will enhance the clarity in Ln 96-Ln 97. Here, 's' denotes the similarity criterion, i.e., cosine similarity. In the literature, visual inputs $v_i$  are typically transformed into features $f(v_i)$ via an encoder. The global features are obtained through global max-pooling or global average-pooling, where the pooling process is not explicitly shown in Eq.(1). We will clarify our writing accordingly.
>
>
> > Q3: Do you use the same (augmented) input and network structure for the two student models? The two models should have individuality and perform similarly to make the consensus meaningful. Are they optimal in this aspect, or is there room for improvement?
>
> * Two students employ different backbones with distinct prior knowledge and pre-training data. Besides, the data fed to the two students consists of different strong augmentation versions of the same instance. These two factors collectively ensure both diversity and individuality between the two models.
> * To ensure that both students approach the correct answer in stage-2 and preventing model collapse, we maintain the utilization of $\mathcal{L}_\text{sup}$ on $\mathcal{D}_l$ and $\mathcal{L}_\text{unsup}$ on $\mathcal{D}_u$. More importantly, the filtering mechanism efficiently sieves out noisy samples and mitigates confirmation bias. Here, we provide additional results on Flickr10k as follows.
>
> | Backbone |  R+R  |  S+S  |  V+V  |  R+S  |  R+V  |    V+S    |
> | :------: | :---: | :---: | :---: | :---: | :---: | :-------: |
> |   CIoU   | 84.0  | 84.8  | 85.2  | 88.6  | 89.2  | **90.4**  |
> |   AUC    | 63.06 | 65.58 | 65.72 | 66.42 | 68.20 | **69.36** |
>
>
> |      | same $\mathcal{B}$, same $\mathcal{A}$ | same $\mathcal{B}$, diff $\mathcal{A}$ | diff $\mathcal{B}$, same $\mathcal{A}$ | diff $\mathcal{B}$, different $\mathcal{A}$ |
> | :--: | :------------------------------------: | :------------------------------------: | :------------------------------------: | :-----------------------------------------: |
> | CIoU |                  84.8                  |                  85.4                  |                  87.2                  |                  **90.4**                   |
> | AUC  |                 65.38                  |                 65.96                  |                 66.28                  |                  **69.36**                  |
>
> where $\mathcal{B}$ and $\mathcal{A}$ denotes 'backbone' and 'augmentation' respectively. Results in the above two tables validate that distinct backbones and augmentations of two models make consensus meaningful and avoid confirmation bias.

---

> ### Author Response · Authors · 2023-08-20
> **Inquiry About Any Additional Concerns?**
>
> Dear Reviewer dVkQ,
>
> We have respond to your concerns carefully and provided relevant experimental results and explanations for each point.
>
> Your insights are highly valuable to us, and we would greatly appreciate your feedback if you have any remaining concerns or suggestions. Please feel free to share with us, as we are more than willing to discuss with you and provide any additional clarifications as needed. Thank you for your consideration.
>
> Looking forward to your early reply.
>
> Sincerely, 11428 Authors.

---

> > ### Comment · Reviewer_dVkQ · 2023-08-22
> >
> > What are the backbones of the three models in "three models ensembling"? Why model 1 has a significantly lower performance (84.4) than others? Are the backbones of DMT S and V? Which gave 89.2?

---

> > > ### Author Response · Authors · 2023-08-22
> > > **Response to Reviewer dVkQ**
> > >
> > > Thank you for reviewing our rebuttal and engaging in discussion with us. We'll address your questions and concerns point by point.
> > >
> > > > Q1: What are the backbones of the three models in "three models ensembling"?
> > >
> > > The three models in the ensemble experiment use VGGish, SoundNet, and ResNet-18 separately as their backbones.
> > >
> > > In the experiments, we trained three models independently in order to fairly compare the distinctions between DMT and ensemble learning. Each of the three models was trained in a semi-supervised mechanism, employing EMA and naive pseudo-labeling (without filtering). During inference, we tested the localization performance of the ensemble by averaging the prediction maps generated by these three models' outputs.
> > >
> > > > Q2: Why model 1 has a significantly lower performance (84.4) than others?
> > >
> > > * Different backbones yield diverse performance due to task suitability, bias influence, overfitting tendencies, and distinct pretraining data. Hence, different models exhibit varied audio-visual source localization performance.
> > >
> > > * Three models are trained independently without interactions. Single models could not implement filtering and IPL, as a result, they are susceptible to confirmation bias, and the results are highly volatile, leading to suboptimal performance randomly.
> > >
> > > * In fact,  there's a likelihood that model2 and model3 could experience severe bias accumulation, resulting in poorer performance. Due to random bias and notable instability, it becomes uncertain that which model will perform better or worse in individual training instances.  This highlights the difference between DMT and ensemble learning, as simple summation can neither overcome instablity nor achieve strong performance.
> > >
> > > > Q3:  Are the backbones of DMT S and V? Which gave 89.2?
> > >
> > > Yes, DMT employs VGGish(V) and SoundNet(S). In this experiment, the CIoU of V and S are 89.2 and 88.8 respectively. It's worth noting that within DMT, two models perform at the same level. Both achieve robust localization with notable stability and minimal variance.
> > >
> > > In general, in DMT, dual teachers interact with each other and jointly select stable samples and assign high-quality pseudo-labels. Consequently, DMT demonstrates both **stability and enhanced performance**, highlighting the clear distinction from naive ensemble learning and showcasing its advantage in audio-visual source localization tasks.
> > >
> > > Finally, we thank the reviewer again and hope our response could address the concerns.

---

### Official Review · Reviewer_13tz · 2023-07-06

**Soundness:** 4 excellent
**Presentation:** 4 excellent
**Contribution:** 4 excellent
**Rating:** 8
**Confidence:** 4

**Summary:**

In this paper the authors address the AVSL task (Audio Video Source Localization), where given an image $v$ and an audio sample $a$, it is required to obtain a confidence map $P$ over $v$, which localizes in $v$ the object heard in $a$. Current approaches adopt self-supervised learning in the form of unsupervised contrastive learning: given a latent embedding $f(v_i)$ of the image divided in patches and a latent embedding of the audio clip $g(a_j)$, embeddings of naturally occurring pairs $(v_i, a_i)$ are brought together, with a loss $L_\text{unsup}$ minimizing the distance between $g(a_j)$ and (a patch of) $f(v_i)$ and maximizing the distance between unrelated pairs. Such models have difficulty localizing small objects and suffer from blurry backgrounds. The authors propose adopting an unbiased semi-supervised learning scheme called Dual Mean Teacher, where a supervised loss $L_\text{sup}$ is additionally employed during the learning phase, matching labeled boundary boxes with the predicted confidence maps $P$. The training proceeds in two stages. During stage one (Warm-Up Stage), they train two teacher AVSL backbones with $L_\text{sup}$, using labeled data and strong augmentation for avoiding overfitting. During stage two (Unbiased-Learning Stage), they train two student networks following the Mean Teacher scheme (https://arxiv.org/abs/1703.01780) (using augmentation and updating the teachers with the EMA of the student weights). The student networks are trained both with $L_\text{unsup}$ and $L_\text{sup}$. For $L_\text{sup}$, the labels are extracted from the labeled dataset or are given as pseudo-labels from the teachers, estimated on the unlabeled dataset. The author proposed the novel idea of using two teachers/students in order to mitigate confirmation bias: i. they perform noise filtering, namely when the two teachers output two pseudo-labels, whose IoU (Intersection over Union) is under a threshold $\tau$, then the input sample is discarded as a false-positive. ii. By intersecting the output pseudo-labels (IPL), one can get a better estimate of the confidence map, and as such, the IPLs are the pseudo-labels fed to the students. Using this training scheme, the authors reach 90.4% CIoU on Flickr 144k and 47.6% CIoU on VGG-SS 144k, beating the competitor models by a large margin.


**Strengths:**

- To the best of my knowledge, the idea of using a pair of teachers/students in semi-supervised learning is novel, naturally realizing the idea of two different teacher models that can correct their estimates to provide better and unbiased pseudo-labels to student models. Such a novelty could be applied not only in the task of AVSL but to all cases in which a metric (IoU in this case) can compare different estimates, so as to remove false positives and where there exists a fusing operation (intersection in this case) that can blend to better estimates.

- The authors successfully adapt semi-supervised learning to a task that has been classically dominated by unsupervised contrastive learning, efficiently integrating the usage of labeled data for predicting audio sources inside images. This is reflected in the strong quantitative results showcased in the experimental section, where consistently the method obtains the highest scores in all benchmarks. Using this training scheme, the authors reach 90.4% CIoU on Flickr 144k and 47.6% CIoU on VGG-SS 144k, beating the competitor models by a large margin.

- Authors showcase qualitative samples that clearly better localize small objects with respect to other baselines.

**Weaknesses:**

-  It is true that the Semi-supervised second stage that integrates all the novel ideas proposed in a paper provides improvements in metrics (e.g., improving from 86.20% to 90.40% CIoU on Flickr 144k), but what I find odd is that already at stage-1, the obtained results beat all the baselines. This may indicate that "supervised is all you need" and that semi-supervised is not the key factor in improving the results (although methodologically the framework is very natural). The paper would definitely benefit from a more detailed scaling study, proving that indeed scaling to more unlabeled data, the semi-supervised mechanism is well exploited.

**Questions:**

- The authors mentioned that Instant-Teaching (https://arxiv.org/abs/2103.11402) uses two networks to correct pseudo-labels. What are the differences between the proposed dual Mean Teacher with the aforementioned method?
 - Given the proposed idea of a dual Mean teacher framework, why the authors preferred showcasing it on the task of AVSL instead of the more broad task of object localization? Is there a reason that such an approach is more naturally applicable to AVSL than to more basic object localization tasks?
 - Is there any relationship between the proposed model and ensemble methods? The idea is that ensemble methods create an ensemble with models that have been trained, and the ensemble is used at inference time. In this case, it is like dynamically ensembling the teacher models (and the students across the EMA updates).

 In the following, I add remarks and other misc questions for the improvement of the paper:

- Line 48: Authors maybe are referring to Figure 2.
 - Line 94: Not being an expert in contrastive learning in AVSL, I had to find the correct reference to better gain an understanding of $L_\text{unsup}$ (https://arxiv.org/pdf/2203.09324.pdf), maybe the authors could put a pointer for the readers in this section?
 - Line 96: Please define $\mathcal{D}_u$ (it is defined later, but it should be here) and $\tau$
 - Line 147: Using $\tau$ again as on Line 96. Should use another letter to not make confusion.
 - Line 158: So if I understand correctly, consistency regularization is applied only on the filtered unlabeled input, and it is computed as the cross entropy between the IPL and the prediction of the students?
 - Line 159: I believe there is a typo because superscripts should be `s`.
 - Line 162: Authors could explain a little bit more in detail the attention module. Does it modulate the visual branch using the current audio-dense prediction?
 - Line 164: $D_u$, which could contain false positives, is still used when training the unsupervised branch?
 - Line 192: Put Appendix number.
 - Line 195: Please better explain what "Considering previous methods often select the first 50% points of the entire image as foreground area of IoU, which is not suitable for small objects," means.
 - Line 199: Appendix in uppercase.
 - Line 204: What is $\sigma$?
 - Line 207: No `:` at the end of the paragraph.
 - Line 210: Remove `when`.
 - Line 242: Why R+R? Shouldn't it be R+V or R+S?
 - Line 274: How do you get 200K if ds is 144k?
 - Line 290: Cannot understand this curve. Higher IoU means that the pseudo-label is better. In such case should filtering improve IoU?
 - Line 308: I don't understand what it means that we can filter when we don't have a filter.

**Limitations:**

The authors do not have limitations sections. The authors should discuss how to improve their method.

---

> ### Author Rebuttal · Authors · 2023-08-09
>
> Thank you for your insightful advice and valuable questions, we will respond to your concerns point by point.
>
> > Q1: "Supervised is all you need". Need more detailed scaling study with more unlabeled data.
>
> * Pure supervised-trained models suffer from severe over-fitting with poor generalization ability, while DMT could circumvent this issue by leveraging more unlabeled data with stable pseudo-labels. In addition to the results in our paper (Tab. 6 and Appendix C.1), we devised more detailed scaling studies as below (CIoU values).
>
> | Multiple  | train & test on VGG | train on VGG & test on Flickr | train & test on Flickr |
> | :------------------------------------------: | :-----------------: | :---------------------------: | :--------------------: |
> |                4k supervised                 |        43.80        |             82.88             |         86.20          |
> |                   (4k/10k)                   |        46.40        |             85.80             |         88.00          |
> |                   (4k/40k)                   |        47.60        |             87.60             |         88.40          |
> |                   (4k/80k)                   |        48.60        |             89.00             |         89.20          |
> |                  (4k/200k)                   |        49.80        |             91.40             |         91.20          |
>
> where values in multiple denote ($|\mathcal{D}_l|/|\mathcal{D}_u|$).
>
> * DMT shows **better generalization ability** than supervised training. From the table above, when the model trained on VGG and test on Flickr, pure supervision suffers from severe degragation compared with the one trained on the same dataset (86.20 drops to 82.88), while DMT maintains stronger generalization performance (86.20 drops to 85.80).
> * DMT more **effectively harnessing data**. Firstly, as the size of unlabeled data grows, DMT obtains consistent performance gains (10k to 200k, 85.80 to 91.40), which proves that DMT could benefit from more unlabeled data due to the stable pseudo-labels.
>
>
> > Q2: What are the differences between the proposed dual Mean Teacher with Instant-Teaching[1].
>
> * Firstly, instant-teaching implement SSL in a co-rectify manner, where one of the two teachers gives pseudo-labels to the other, while dual teachers in DMT collectively produce IPL for pseudo-labeling.
> * Secondly, there is no explicit instance-level noisy filtering mechanism in instant-teaching, while DMT filters noisy samples via consensus and only assign pseudo-labels to stable samples.
>
>
> > Q3: How about apply DMT to more broad task of object localization? Is there a reason that such an approach is more naturally applicable to AVSL?
>
> * In this paper, we focus on the novel AVSL where numerous challenges still persist, e.g, small objects localization, poor localization performance. The initial motivation of consensus filtering is to bypass false positives, which could also be use to filter other types of noisy samples.
> * The core spirit of DMT is valuable and generalizable, we will apply DMT to more general spatial (object detection) and temporal (event localization) localization tasks in the future.
>
>
> > Q4: Is there any relationship between the proposed model and ensemble methods?
>
> * Many existing ensemble learning techniques, such as bagging, random forest, primarily focus on ensemble predictions. These methods involve training multiple models independently and simply combining their predictions during testing.
> * The proposed dual teachers interactly with each other and collectively avoid confirmation bias by sample filtering and pseudo-labels, they influence each other instead of trained independently.
> * DMT is irrelevant to ensembling learning, there is interaction between dual teachers in training and inference, not simply ensembling the results or implement voting.
> * Also, refer to the Table in **response to Q1 of reviewer dVkQ**, DMT shows advantages over naive ensembling.
>
>
> > Q5: misc questions.
>
> * Ln. 48,94,96,147,159, 192 are typos, we will fix them.
> * Ln 158: Consistency regularization is applied on both the labeled and unlabeled data. In stage-2, we employ ground-truth label for labeled data and IPL for filtered unlabeled data. In both cases, students receive images of strong augmentation images while teachers with weak ones. Loss functions are cross entropy between the ground-truth labels (for $\mathcal{D}_l$) or IPL (for $\mathcal{D}_u^\prime$) and the prediction of the students.
> * Ln 164: For $\mathcal{L}_\text{unsup}$, we also use filtered unlabeled data $\mathcal{D}_u^\prime$ to avoid false positives. Over the course of training, only $\mathcal{D}_u^\prime$ and $\mathcal{D}_l$ are sent to the model. We will clarify it in the revised version.
> * Ln 195: In the literature of AVSL, among all the pixels, the pixel with the top 50% of the prediction confidence is defined as the foreground, then the foreground is employed to compute IoU with ground-truth bounding box.
> * Ln 204: It is a typo, here we mean $\delta$.
> * Ln 242: We add 'R+V' and 'R+S' results, see below.
>
> |              |  R+V  |  R+S  |
> | :----------: | :---: | :---: |
> | EZVSL w/ DMT | 85.30 | 85.94 |
> | SLAVC w/ DMT | 86.10 | 86.30 |
>
>
> * Ln 290: Sorry, in Fig. 4 (a), there is a typo, the blue curve is 'After Filtering' while the red denotes 'Before Filtering'. Fig. 4 (a) proves that filtering module could impove the quality of pseudo-labels. We will fix the legend.
> * Ln 308: When there is no filter, the filtered numbers degenerate to the total numbers of the unlabeled data. We will refine it.
>
>
> > Q6: The authors should discuss limitations and how to improve their method.
>
> We add the limitations, please refer to **General Response**.
>
>
> **References:**
> [1]. Zhou Q, Yu C, Wang Z, et al. Instant-teaching: An end-to-end semi-supervised object detection framework. CVPR, 2021.

---

### Official Review · Reviewer_GZse · 2023-07-17

**Soundness:** 3 good
**Presentation:** 3 good
**Contribution:** 2 fair
**Rating:** 5
**Confidence:** 5

**Summary:**

In this paper, the authors present a Semi-Supervised Learning framework, Dual Mean-Teacher (DMT), for Audio-Visual Source Localization (AVSL). DMT improves object localization in video frames paired with audio clips by employing two teacher-student structures to mitigate confirmation bias and generate high-quality pseudo-labels. The method notably improves AVSL performance, surpassing other methods on Flickr-SoundNet and VGG-Sound Source with only 3% of data being positionally annotated. The framework can also enhance existing AVSL methods.

**Strengths:**

+ The paper is easy to follow, and the method section is clear.

+ The proposed semi-supervised method outperforms past unsupervised approaches and can be extended to boost different AVSL methods.

**Weaknesses:**

+ Semi-Supervised Audio-Visual Source Localization is not new. Back in 2018, Senocak et al. had already extended a semi-supervised approach to leverage labeled data to help with audio-visual source localization. I was surprised that the authors did not mention this in the paper. The experiments in the paper have larger benefited from the insight, experimental setting, and evaluation of the pioneering work.

+ The effectiveness of the proposed Dual Mean-Teacher (DMT) is not fully validated. (1) The authors did not compare with existing semi-supervised AVSL methods (e.g., [1, 2]). (2) Existing semi-supervised learning approaches can be applied to solve AVSL. The authors should also add these naive A+B baselines into comparison. Without these comparisons, we cannot understand the significance of the performance improvements. For instance, in [2], the very weak Attention10k model with a naive semi-supervised learning trained on Flickr 10k can achieve 84% cIOU. The proposed method with a strong model and an advanced semi-supervised method under a similar setting achieves 87.80%.

+ The technical contributions are limited. As stated in the paper, the key body of the used semi-supervised learning method is from existing work. The contributions of Warm-Up and Noise Filtering tricks are marginal.

+ The demo only shows localization results on one example and did not compare with other approaches.

+ How does the selection of teacher models affect AVSL performance? I think it is important to ensure that the two teachers produce accurate but different pseudo-labels. More experiments and analyses on teacher selection should be provided.

 [1] Arda Senocak, Tae-Hyun Oh, Junsik Kim, Ming-Hsuan Yang, and In So Kweon. Learning to localize sound source in visual scenes. CVPR, 2018.

 [2] Arda Senocak, Tae-Hyun Oh, Junsik Kim, Ming-Hsuan Yang, and In So Kweon. Learning to Localize Sound Sources
in Visual Scenes: Analysis and Applications. TPAMI, 2019.

**Questions:**

See Weaknesses.

**Limitations:**

The authors did not discuss method limitations in the paper.

---

> ### Author Rebuttal · Authors · 2023-08-09
>
> Thank you for your insightful advice and valuable questions, we will respond to your concerns point by point.
>
> > Q1: Semi-Supervised Learning (SSL) in AVSL is not new. Relationship to existing methods (Attention10k[1,2]).
>
> * Thank you for your suggestions. We have already cited and compared the unsupervised methods from [1,2] in the main paper, and will incorporate the semi-supervised results in [1,2] into the revised paper.
> * Pioneering works [1,2] introduced SSL to AVSL. However, the motivation, idea and contribution of DMT are novel and distinct from [1,2]. Please refer to **General Response**.
> * We summarize the core differences between DMT and Attention10k[1,2].
>   * Regarding SSL scheme, [1,2] naively employed $\mathcal{L}_\textrm{sup}$ on $\mathcal{D}_l$, without utilizing pseudo-labels from $\mathcal{D}_u$. In contrast, DMT introduced the pseudo-label mechanism to make full use of $\mathcal{D}_u$ and addressed the confirmation bias issue.
>   * The dual-stram network in [1,2] serves as an unsupervised method, which is orthogonal to the proposed DMT framework, one could combine them to further boost performance.
>
>
> > Q2: The experiments in the paper have larger benefited from the insight, experimental setting, and evaluation of the pioneering work.
>
> We adopted common setups for fair comparisons. We have also introduced our distinct settings. For instance, we introduced a new metric, MSE (Sec. 5.1, line 197), to validate small object localization. We also assess the quality of pseudo-labels quantitively (Fig. 4) and conducted experiments with various label ratios (Tab. 6).
>
>
> > Q3: The effectiveness of the proposed Dual Mean-Teacher is not fully validated. The authors should also add naive A+B baselines into comparison.
>
> * We appreciate your constructive advice. We devise the naive A+B experiments for Attention10k for more comprehensive comparisons. Here, 'naive SSL' means only employing $\mathcal{L}_\text{sup}$ on $\mathcal{D}\_l$ apart from $\mathcal{L}\_\text{unsup}$ without any pseudo-labels. Results (CIoU) are shown below.
>
> |             method              | sup1k+unsup10k | suo2.5k+unsup10k | sup2.5k+unsup144k |
> | :-----------------------------: | :------------: | :--------------: | :---------------: |
> |          Attention10k           |      43.6      |       43.6       |       66.0        |
> |     Attention10k+naive SSL      |      82.4      |       84.0       |       84.4        |
> |              SLAVC              |      66.8      |       66.8       |       73.8        |
> |         SLAVC+naive SSL         |      80.4      |       82.2       |       83.8        |
> |          unsup (ours)           |      65.6      |       65.6       |       72.4        |
> |     unsup (ours)+naive SSL      |      81.2      |       82.8       |       83.6        |
> | unsup (ours)+naive pseudo label |      80.8      |       81.6       |       81.8        |
> |           DMT (ours)            |    **87.4**    |     **88.0**     |     **89.6**      |
>
> * **label ratio.** Results show that SSL could improve performance, and DMT surpasses naive SSL as $|\mathcal{D}_u|$ increases. The proposed IPL enhances DMT with less confirmation bias and more utility of $\mathcal{D}_u$.
> * **Generalization.** We compare naive SSL and DMT in terms of generalization ability to unseen objects, i.e., open set and cross-dataset results, as in **Rebuttal PDF**.
> * The results above highlight the validity of DMT and its advantages over naive SSL methods. We will incorporate these in the revised paper and integrate discussions concerning[1,2].
>
>
> > Q4: The technical contributions are limited. The contributions of Warm-Up and Noise Filtering tricks are marginal.
>
> * The central motivation lies in addressing the confirmation bias in SSL and achieving more efficient data utilization, which is under-explored in AVSL. DMT introduces a novel dual teacher-student architecture, harnessing the consensus of two teachers to rectify and generate high-quality pseudo-labels. DMT is a learning paradigm characterized by robust transferability and adaptability.
> * **Warm-Up** helps to avoid incorrect pseudo-labels from two teachers at the beginning and circumvent error accumulation[3]. We have empirically validated the necessity of warm-up in Sec. 5.4, Fig. 3, and Appendix C.3. **Filtering** is an online training method instead of simple data cleaning.
> * Ablations (Tab. 5) show that the absence of any module leads to significant performance degradation, e.g., without IPL, CIoU drops from 90.40 to 86.20, and without the filter, it decreases to 86.60.
>
>
> > Q5: The demo did not compare with other approaches.
>
> We provide more comparison demos and the link is sent to AC as per the conference requirements.
>
>
> > Q6: How does the selection of teacher models affect AVSL performance?
>
> * The key to the success of dual teachers is that the two models are individual and will not converge to the same. Hence, we select distinct backbones and apply diverse augmentations to the two models, ensuring their independence.  In Table 4, we extend DMT to SLAVC and EZVSL, diverse backbones surpassed identical backbones. For instance, in EZVSL, (V+S) CIoU (87.20) outperforms (R+R) CIoU (82.80).
> * We have also added experiments with more combinations, please refer to the Table in **Response to Q3 of Reviewer dVkQ**.
>
>
> > Q7: The authors did not discuss method limitations in the paper.
>
> We list the limitations in **General Response** and will add them to the revised paper.
>
>
> **References:**
> [1]. Arda Senocak, Tae-Hyun Oh, Junsik Kim, Ming-Hsuan Yang, and In So Kweon. Learning to localize sound source in visual scenes. CVPR, 2018.
> [2]. Arda Senocak, Tae-Hyun Oh, Junsik Kim, Ming-Hsuan Yang, and In So Kweon. Learning to Localize Sound Sources in Visual Scenes: Analysis and Applications. TPAMI, 2019.
> [3]. Arazo E, Ortego D, Albert P, et al. Pseudo-labeling and confirmation bias in deep semi-supervised learning. IJCNN, 2020.

---

> > ### Comment · Reviewer_GZse · 2023-08-16
> > **Post-rebuttal**
> >
> > I appreciate the authors' response to my questions. However, I believe that the rebuttal does not fully address my major concerns.
> >
> > Considering the existing works in SSL for AVSL, I do not think the motivation “The existing methods struggle to effectively utilize the abundance of unlabeled audio-visual pairs alongside the limited yet valuable annotated data" is still fully valid. I was super surprised that the authors did not mention the existing works that can use SSL for AVSL especially considering this work adopts dataset and experimental setting in the pioneering works.
> >
> > Regarding the new results presented in the rebuttal, I am not convinced of the effectiveness of the proposed method. First, the model used in this work is much stronger than Attention10k. Therefore, the improvements over Attention10k+naive SSL do not demonstrate the effectiveness of the approach. Second, the unsup (ours)+naive SSL results are even worse than Attention10k+naive SSL. These results are clearly not convincing.
> >
> > I will keep my previous rating (Reject).

---

> > > ### Author Response · Authors · 2023-08-19
> > > **Response to Reviewer GZse [Part 1/2]**
> > >
> > > We appreciate the reviewer's consideration of our rebuttal and thoughtful response and we will address the concerns left by the reviewer thoroughly.
> > >
> > > > Q1: Should cite and compare with naive SSL.
> > >
> > > We acknowledge the reviewer's perspective and agree this work should be discussed and compared with. Previously, we acknowledged the novelty of Attention10k for introducing a two-stream network structure in self-supervised learning, where SSL entails a supervised loss component, while our novelty lies in an unbiased SSL framework based on pseudo-label. Given this gap, we omitted naive SSL as comparison.
> > >
> > > We have added proper discussion and experimental comparison in our manuscript.
> > >
> > > **Introduction(Ln36-43)**:
> > >
> > > > Considering that unsupervised AVSL is not fully learnable, Attention10k[1,2] extended the unsupervised model to semi-supervised model by directly appending a supervised loss on labeled data, which is the first semi-supervised AVSL work. However, simply leveraging labeled data might lead to overfitting and neglect to fully harness the underlying unlabeled data. Given this, we resort to pseudo-labeling. However, directly introducing pseudo-labeling could lead to *confirmation bias* which cannot be adequately rectified by a single model[12]. To address the issue, we advance the naive SSL to a more sophisticated framework, called Dual Mean-Teacher (DMT).
> > >
> > > **Related Works(Ln82:)**:
> > >
> > > > Attention10k [1,2] is the first SS-AVSL work. It extends an unsupervised model to a semi-supervised framework by simply adding a supervised loss, aiming at fixing the false conclusions generated by unsupervised methods. However, this naive method may lead to overfitting and neglects the full utilization of unlabeled data. In contrast, we introduce a novel SS-AVSL framework based on pseudo-label mechanism, which can address confirmation bias and maximize the utilization of both labeled and unlabeled data, to achieve stronger localization performance, with better small object localization, handling off-screen issue and strong generalization capability.
> > >
> > > **Conclusion**:
> > > > Ln 311: In this paper, we advance the naive SS-AVSL work Attention10k [1,2] and propose a novel SSL framework called Dual Mean-Teacher.
> > >
> > > > Ln 319: We hope this work will bring more attention to SS-AVSL, provoke a reconsideration of pseudo-labeling and bias avoidance, and better utilization of the underlying unlabeled data.
> > >
> > > We hope the reviewer could acknowledge these revisions.
> > >
> > > ---
> > > > Q2: About the motivation "The existing methods struggle to effectively utilize the abundance of unlabeled audio-visual pairs alongside the limited yet valuable annotated data".
> > >
> > > We revise it to reduce possible misunderstandings: "Existing methods have not fully leveraged the information from both labeled and unlabeled data, leading to suboptimal performance in certain cases, like generalization capability."
> > >
> > > In fact, this motivation stems from the two points:
> > > * Existing unsupervised methods solely use unlabeled data without any positional labeled data.
> > > * Existing semi-supervised method naive SSL simply adding a supervised loss might lead to overfitting and neglect to fully harness underlying unlabeled data.
> > >
> > > Both naive SSL and DMT have utilized labeled and unlabeled data. However, a key distinction is that naive SSL employs unlabeled data only for unsupervised loss, whereas DMT leverages pseudo-labels to incorporate unlabeled data into both unsupervised and supervised loss, which amplifies the utilization of unlabeled data, thus enhances generalization capability.
> > >
> > > **Data Utilization**
> > > We supplement the comprision experiments with fixed labeled data and an increase in unlabeled data from 10k to 200k.  (* denotes the results from the original paper, other results are the averages of five experimental runs. 'sim-unsup' denotes the simple unsup model we use)
> > >
> > > |                         |   2.5k/10k    |   2.5k/144k   | 2.5k/200k |
> > > | :---------------------: | :-----------: | :-----------: | :-------: |
> > > | attention10k+naive SSL  | 84.00*/83.68 | 84.40*/84.08 |   84.24   |
> > > | attention10k+DMT (ours) |   **88.00**   |   **89.52**   | **90.40** |
> > > |     sim-unsup+naive SSL |     83.84     |     84.24     |   84.40   |
> > > |    sim-unsup+DMT (ours) |   **88.24**   |   **89.76**   | **91.12** |
> > >
> > > As the amount of unlabeled data increases, naive SSL exhibits only marginal improvement, whereas DMT shows more performance gains, indicating DMT can better use unlabeled data. Exp. in **Q1 of reviewer 13tz** also draws a similar conclusion.
> > >
> > > **Generalization Ability**
> > > Previous rebuttal (Tab.1 in Rebuttal PDF) highlighted the limitations of naive SSL in open set and in-the-wild datasets, suggesting that adding a supervised loss alone may lead to overfitting and weaken generalization. In contrast, DMT effectively leverages pseudo-label for improved generalization capability.
> > >
> > > **We attribute these two superiorty to the effective utilization of unlabeled data through the pseudo-label mechanism in DMT.**

---

> > > > ### Author Response · Authors · 2023-08-19
> > > > **Response to Reviewer GZse [Part2/2]**
> > > >
> > > > > Q3: The model used in this work is much stronger than Attention10k. Therefore, the improvements over Attention10k+naive SSL do not demonstrate the effectiveness of the approach.
> > > >
> > > > In order to highlight the effectiveness of DMT, we compare the results of various $\mathcal{L}_\text{unsup}$ combined with three *SSL frameworks*: **1. naive SSL**, **2. naive pseudo-labeling (naive PL, without filtering)**, and **3. DMT (IPL+filtering)**.
> > > >
> > > > We conduct experiments on Flickr (2.5k/144k) and report CIoU value. (* denotes the results from the original paper, other results are the averages over five runs. 'sim-unsup' denotes the simple unsup model we use.)
> > > >
> > > > |                 | Attention10k | sim-unsup |   SLAVC   |    ETI    |
> > > > | :-------------: | :----------: | :-------: | :-------: | :-------: |
> > > > |      unsup      | 66.00*/65.52 |   72.48   |   73.84   |   81.50   |
> > > > | unsup+naive SSL | 84.40*/84.08 |   84.16   |   84.40   |   84.64   |
> > > > | unsup+naive PL  |    80.32     |   81.80   |   82.60   |   83.44   |
> > > > |    unsup+DMT    |  **89.28**   | **89.76** | **89.92** | **90.24** |
> > > >
> > > >
> > > > Results indicate that both naive SSL and DMT significantly improve the performance of unsupervised methods.
> > > > * **The performance disparities among several $\mathcal{L}_\text{unsup}$ is negligible** when combined with SSL frameworks. Therefore, the gap between the two unsupervised models with naive SSL will not be substantial, and the results will likely be comparable.
> > > > * **Naive pseudo-labeling performs worse than naive SSL** due to the noisy samples and low-quality pseudo-labels, which indicates that the avoidance of confirmation bias in DMT is essential.
> > > > * **DMT surpasses naive SSL**, validating the effectiveness of the framework and our advanced pseudo-labeling mechanism. DMT more fully exploits the potential of the data.
> > > > * **DMT outperforms naive pseudo-labeling** by a large margin, which means that DMT could effectively circumvent confirmation bias and potential noise.
> > > >
> > > > **In conclusion, the results above validate the effectiveness of DMT.**
> > > >
> > > >
> > > > > Q4: The unsup+naive SSL results are even worse than Attention10k+naive SSL.
> > > >
> > > > In this table, we reported the official results of attention10k and attention10k+naive SSL [1,2]. There will be differences between the results in the paper and other methods with different experimental settings (such as architecture, batch size, frame size) and environment compared to other methods.
> > > >
> > > > We consider the observed discrepancy in results to fall within an acceptable range of error,due to variations in settings, environments and random seeds, and it does not undermine the effectiveness of both two methods.
> > > >
> > > > To ensure a fair comparison within a consistent experimental environment, we replicated the attention10k and naive SSL experiments using the official codebase. Our reproduced results show a slight decrease(within 1%) in performance, as similarly indicated in the original GitHub repo of Attention10k[1,2] ("Accuracy (with PyTorch) is slightly lower than the reported number in the paper (with Tensorflow)").
> > > >
> > > > Moreover, **we average results from five experimental runs, as shown in the table above,** in order to avoid mitigate errors in our experiments.
> > > >
> > > >
> > > > ---
> > > > If you still have any concerns, please discuss them with us. Looking forward to your reply.
> > > >
> > > > **References:**
> > > >
> > > > [1]. Arda Senocak, Tae-Hyun Oh, Junsik Kim, Ming-Hsuan Yang, and In So Kweon. Learning to localize sound source in visual scenes. CVPR, 2018.
> > > >
> > > > [2]. Arda Senocak, Tae-Hyun Oh, Junsik Kim, Ming-Hsuan Yang, and In So Kweon. Learning to Localize Sound Sources in Visual Scenes: Analysis and Applications. TPAMI, 2019.

---

> > > > > ### Comment · Reviewer_GZse · 2023-08-19
> > > > > **Response to Authors**
> > > > >
> > > > > I appreciate the new response from the authors. The modified introduction and related works, as well as the additional details and results, have clarified my major concerns. I would be happy to recommend the acceptance of this paper if the authors incorporate these modifications into the final version.
> > > > >
> > > > > I have upgraded my rating.

---

### Author Rebuttal · Authors · 2023-08-09

We thank all reviewers for their dedication to our paper and insightful comments, and we believe these comments are significant for improving the overall quality of this paper.

We are pleased that the reviewers appreciate our paper from various aspects, including its strong performance [GZse] [13tz] [dVkQ], novelty [13tz], scalability [GZse], comprehensive experiments [1pQt] [13tz], clear and concise writing [GZse] [MLW4], and reproducibility [GZse] [MLW4]. These positive assessments truly motivate us.

The **motivation** of DMT is to solve the following problems:
*  The existing methods struggle to effectively utilize the abundance of unlabeled audio-visual pairs alongside the limited yet valuable annotated data.
* In semi-supervised learning, the issue of confirmation bias caused by low-quality pseudo-labels significantly undermines the training performance of the model.

Thus, the **core idea** of Dual Mean-Tacher (DMT) is to introduce a dual teacher-student structure, leveraging consensus from two teachers, incorporating a filtering mechanism, and utilizing the Intersection of Pseudo-Labels (IPL) module. This approach can rectifie and generate high-quality pseudo-labels online, effectively addressing the issue of confirmation bias.

We summarize our **contributions** as follows:
* We introduce a novel unbiased framework based on pseudo-label mechanism for semi-supervised AVSL, which could maximize the utilization of both labeled and unlabeled data, effectively address the challenge of limited annotated data and mitigates the issue of confirmation bias.
* Compared to existing approaches, DMT achieves much stronger localization performance, with better small object localization, handling off-screen issue and strong generalization capability. These strengths significantly elevate the performance of current weakly-supervised (previously referred to as 'unsupervised') methods.
* DMT can be summarized as a semi-supervised learning paradigm and could be combined with existing (weakly-supervised) AVSL methods to consistently boost their performance.
* The spirit of DMT could also be applied to other broad localization tasks.

The **limitations** of DMT are as follows:
* DMT does not involve class information, so it struggles to localize among fine-grained objects due to poor discriminative ability. By incorporating category signals, models could better implement fine localization. Besides, future work could consider employing huge pre-trained models, e.g. ViT pre-trained with DINO.
* DMT could not handle multi-objects localization well. We will devise specialized components to address this issue.

In this rebuttal, we have added **more supporting results** following the reviewers’ suggestions.

* comprehensive comparisons with supervised, existing SSL methods and naive A+B SSL baseline methods [GZse,13tz]
* scaling experiments with various unlabeled data [13tz]
* comprison ensemble learning baseline [dVkQ]
* model collapse analysis [MLW4]
* results on diverse domains including MUSIC and audioset [1pQt,MLW4]
* more visualization and ablation results on the proposed modules in DMT [MLW4,dVkQ]

---

### Author Response · Authors · 2023-08-21
**Rebuttal Summary and Appreciation to the Reviewers**

We sincerely appreciate the dedication and constructive suggestions from all the reviewers. We are grateful for the recommended acceptance from them and find great encouragement in their acknowledgment. Our paper has been significantly improved during this discussion phase.

In general, our work has made substantial progress in 1). how to effectively utilize labeled and especially unlabeled data and 2). how to mitigate confirmation bias in localization. The proposed DMT, which relies on a novel pseudo-labeling mechanism, has tackled challenges in Audio-Visual Source Localization (AVSL) and showcased remarkable performance improvements, particularly in scenarios involving small objects and off-screen sounds. Furthermore, DMT can be summarized as a learning paradigm to tackle confirmation bias issues, and its intersection-based approach could help implement noise filtering and provide inspiration and benefits for various problems.

Undoubtedly, the reviewers have made great contributions to the improvement of our paper. With their guidance, we conducted a comprehensive analysis to explore the capabilities of DMT from a broader perspective. We compared DMT with naive SSL, highlighting the effectiveness of DMT in terms of performance improvement, efficient data utilization, and mitigation of confirmation bias. Moreover, we thoroughly investigated the conditions under which two teachers work effectively, finding that different backbones, augmentations, and annotations in stage-2 ensure the effectiveness of teachers' consensus and prevent model collapse. Additional experiments on diverse datasets from different domains validated our model's generalization capabilities, which provided more insights into its performance. In our final revision, we will incorporate relevant explanations and experimental results as suggested by the reviewers.

We deeply appreciate the recognition of our contributions by the reviewers, and we also hope that our work can draw more attention to this field and inspire more researchers to actively engage in NeurIPS.

---

### Decision · Program_Chairs · 2023-09-21

**Decision:**

Accept (poster)

**Comment:**

The paper received mixed reviews initially. After rebuttal and discussion, the final consensus by all reviewers is to accept the work with the reviewers all being convinced and being positive (ranging from borderline accept to strong accept). The additional results and comparisons provided during rebuttal were quite useful in convincing the reviewers. It is recommended that the paper be accepted. It is strongly recommended to include the discussion and points provided during rebuttal in the final version of the paper.